# Mimicking on-water surface synthesis through micellar interfaces

Anupam Prasoon [1,2], Shaik Ghouse[1], Nguyen Ngan Nguyen [1,2], Hyejung Yang[1], Alina Müller [1], Chandrasekhar Naisa[1,2], Silvia Paasch[1], Abdallh Herbawe [3], Muhannad Al Aiti [3], Gianaurelio Cuniberti [3], Eike Brunner[1] & Xinliang Feng [1,2] ✉

The chemistry of the on-water surface, characterized by enhanced reactivity, distinct selectivity, and confined reaction geometry, offers significant potential for chemical and materials syntheses. However, the utilization of on-water surface synthesis is currently limited by the requirement for a stable air-water interface, which restricts its broader synthetic applications. In this work, we present a approach that mimics on-water surface chemistry using micelles. This method involves the self-assembly of charged surfactant molecules beyond their critical micelle concentration (CMC), forming micellar structures that simulate the air-water interface. This creates an environment conducive to chemical reactions, featuring a hydrophobic core and surrounding water layer. Utilizing such mimicking on-water surface with the assembly of porphyrin-based monomers featuring distinct confined geometry and preferential orientations, we achieve reactivity and selectivity (≥99%) in fourteen different reversible and irreversible chemical reactions. Extending the versatility of this approach, we further demonstrate its applicability to two-dimensional (2D) polymerization on micellar interfaces, successfully achieving the aqueous synthesis of crystalline 2D polymer thin layers. This strategy significantly broadens the accessibility of on-water surface chemistry for a wide range of chemical syntheses.

The on-water surface is ubiquitous in nature and serves as a platform where many fundamental chemical, biological, and environmental processes occur[1]. Chemical reactions on the water surface with confined geometry exhibit enhanced reaction rates, distinct reactivity, and specific selectivity, unlike those observed in bulk phases[2–9]. The pioneering work by Breslow's group in 1980 on the role of water as a solvent (the so-called solution-phase hydrophobic effect) revealed that water could significantly accelerate Diels−Alder reactions, even with nonpolar compounds, thereby paving the way for further exploration into the role of water in chemical synthesis[10]. A significant breakthrough in understanding the on-water effect

emerged in 2005 when Sharpless and his colleagues demonstrated that a series of organic reactions, including $[2\sigma + 2\sigma + 2\pi]$ cycloaddition, Diels−Alder reaction, and Claisen rearrangement, occurred more rapidly when insoluble reactants were stirred in an aqueous suspension than when these reactions were performed in conventional organic solvents[2]. The "on-water effect" was proposed to play a crucial role in catalyzing reactions by influencing the reaction energy barrier[3,5,7,11–14]. However, the specific role of the actual surface (air−water interface, also designated as the on-water surface) in achieving these chemical reactivities has not been fully illustrated due to the lack of a "stable" water surface[15]. Very recently, the

[1]Center for Advancing Electronics Dresden (cfaed) and Faculty of Chemistry and Food Chemistry, Technische Universität Dresden, 01062 Dresden, Germany. [2]Max Planck Institute for Microstructure Physics, Halle (Saale) D-06120, Germany. [3]Institute for Materials Science and Max Bergmann Center of Biomaterials, Technische Universität Dresden, 01062 Dresden, Germany. ✉e-mail: xinliang.feng@tu-dresden.de

development of ultrafast phase-sensitive interface-selective non-linear vibrational spectroscopy, particularly sum-frequency generation (SFG) spectroscopy, has facilitated the study of the characteristics of the water surface in terms of confined molecular geometry/orientation and hydrogen bonding configuration (free OH groups, often referred to as "dangling bonds")[16,17]. SFG spectroscopy selectively enhances interfacial signals while suppressing bulk aqueous phase contributions. The molecular orientation significantly affects interfacial properties such as hydrophobicity and chemical reactivity[18–29]. Additionally, the behavior of these molecular orientations is sensitive to chemical environmental changes (e.g., pH, ionic strength, temperature), allowing for the study of dynamic processes and interactions at the air-water interface[17,30,31]. This leads to a profound understanding of the factors driving the reactivity of the on-water surface[22,23,25].

In the last few years, our group has demonstrated the on-water surface synthesis over large areas (hundreds of cm²) using Langmuir-−Blodgett (LB)[32] and surfactant-monolayer-assisted interfacial

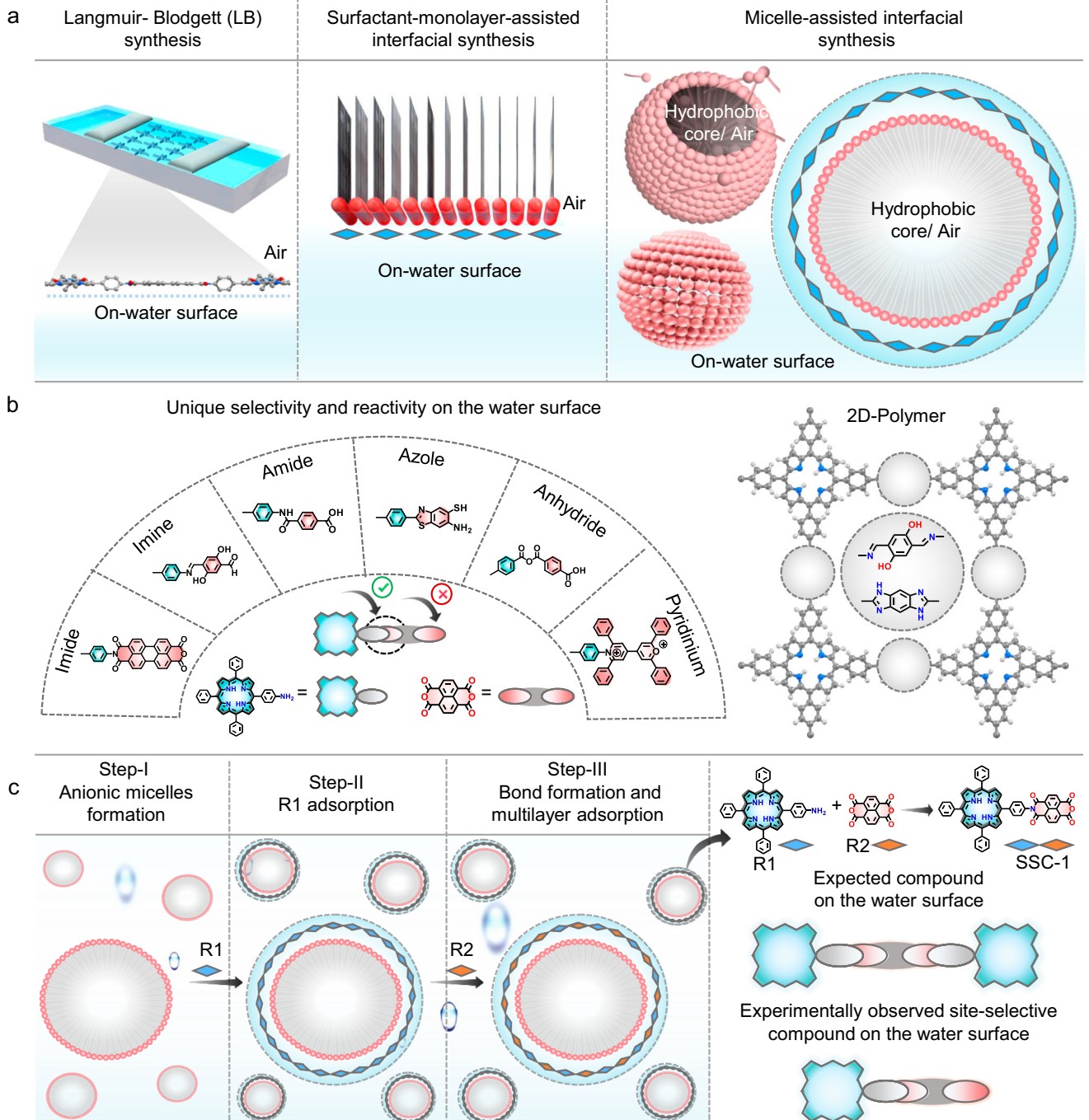

**Fig. 1 | Micellar interfaces mimicking on-water surface synthesis. a** Schematic representation of on-water surface synthesis methods: Langmuir−Blodgett synthesis, surfactant monolayer-assisted interfacial synthesis (SMAIS), and our newly designed micelle-assisted interfacial synthesis. Highlighting the hydrophobic core and the surrounding head groups in contact with water molecules. **b** Exploration of a series of different reactant molecules in a wide range of site-selective reactions and 2D polymer synthesis. **c** A schematic illustration of the step-wise synthesis and experimentally observed site-selective compounds compared to expected compounds on the surface of micelles in an aqueous solution.

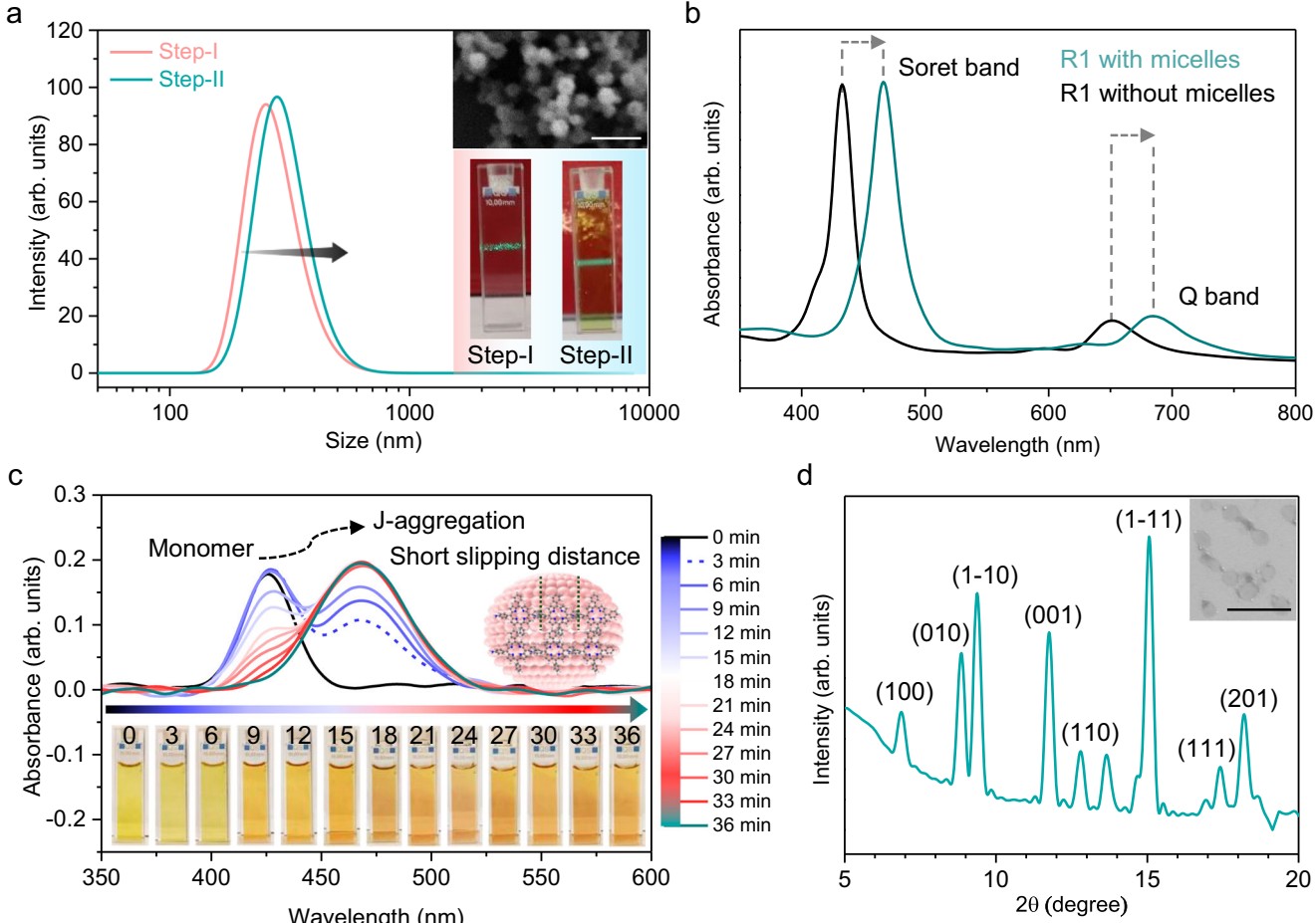

**Fig. 2 | Pre-assembly of J-aggregated R1 on the micelle surface. a** Number-average size of DLS-measured SOS micelles (Step-I) and the assembled structure of SOS-R1 micelles (Step-II). Inset: Photographs of Step-I and Step-II demonstrating the Tyndall effect, with FE-SEM images showing the spherical morphology of the SOS-R1 assembled structure (before washing with THF), with a scale bar of 1 μm. **b** UV–Vis spectra of R1 in the presence and absence of micelles (Step-II). **c** In-situ UV–Vis spectroscopy studies of Step-II (in water), illustrating the short-slip distance in the J-aggregated R1 structure. **d** PXRD analysis of the R1 assembled structure. Inset: FE-SEM images reveal circular sheet-like structures of the SOS-R1 (after washing with THF), with a scale bar of 1 μm.

synthesis (SMAIS)[33–39]. The sequential assembly of reactant molecules on the stable water surface has exhibited reactivity driven by confined geometries and the preferential orientation of the reacting molecules in the form of monolayer to few-layers[39]. Furthermore, on-water surface chemistry has been extended to polymer chemistry, facilitating the precise synthesis of sheet-like polymers with long-range order in two distinct directions, namely two-dimensional (2D) polymers and their layer-stacked 2D covalent organic frameworks (COFs)[40]. Despite these significant advancements, a crucial challenge in on-water surface synthesis using LB or SMAIS remains its limited accessibility, which is currently constrained to a stable air–water interface. This limitation restricts the reactivity and spatial geometry to confined spaces, thereby hindering the development of broader synthetic applications. Addressing this challenge is imperative to fully realize the potential of on-water surface chemistry in diverse and scalable chemical and materials syntheses.

In this study, we present an approach to create an interface, where the chemical reactivity and selectivity are akin to those observed in on-water surface chemistry. By employing the self-assembly of charged (anionic/cationic) surfactant molecules beyond their critical micelle concentration (CMC), resulting in the formation of micellar supramolecular structures. Within these micelles, the hydrophobic cores mimic an air-like environment, while the water layer surrounding the surfactant head groups serves as an interface,

reflecting the conditions found in on-water surface chemistry. The amino-substituted porphyrin molecules pre-organize on the micelle surface into a J-aggregated packing structure, facilitated by the charged head groups of the micelles. This distinctive confined geometry with preferential orientations leads to reactivity and selectivity on the micelle surface. We employ both in-situ and ex-situ techniques to monitor the self-assembly kinetics and surface charge dynamics facilitating multilayer growth of site-selective imide products on micelle surfaces. This approach is demonstrated with exceptional selectivity (≥99%) and excellent yield (≥92%) across various reversible and irreversible chemical reactions involving imide, imine, amide, azole (including thiazole, imidazole, and oxazole), pyridinium, and anhydride bonds. Moreover, we further show the successful aqueous synthesis of crystalline 2D polymer thin layers through the controlled, stepwise sequential assembly of porphyrin-based monomers on the micelle surface. Specifically, we prepared 2D polyimine (2DPI) using the anionic surfactant sodium oleyl sulfate (SOS) and 2D polybenzimidazole (2DPBI) using the cationic surfactant cetyltrimethylammonium bromide (CTAB), yielding uniform circular sheets of ∽2.0 μm for 2DPI and 1.0 μm for 2DPBI, with thicknesses of around 15 and 18 nm, respectively. High-resolution transmission electron microscopy (HR-TEM) reveals highly crystalline structures with well-defined square lattices for both 2DPI and 2DPBI.

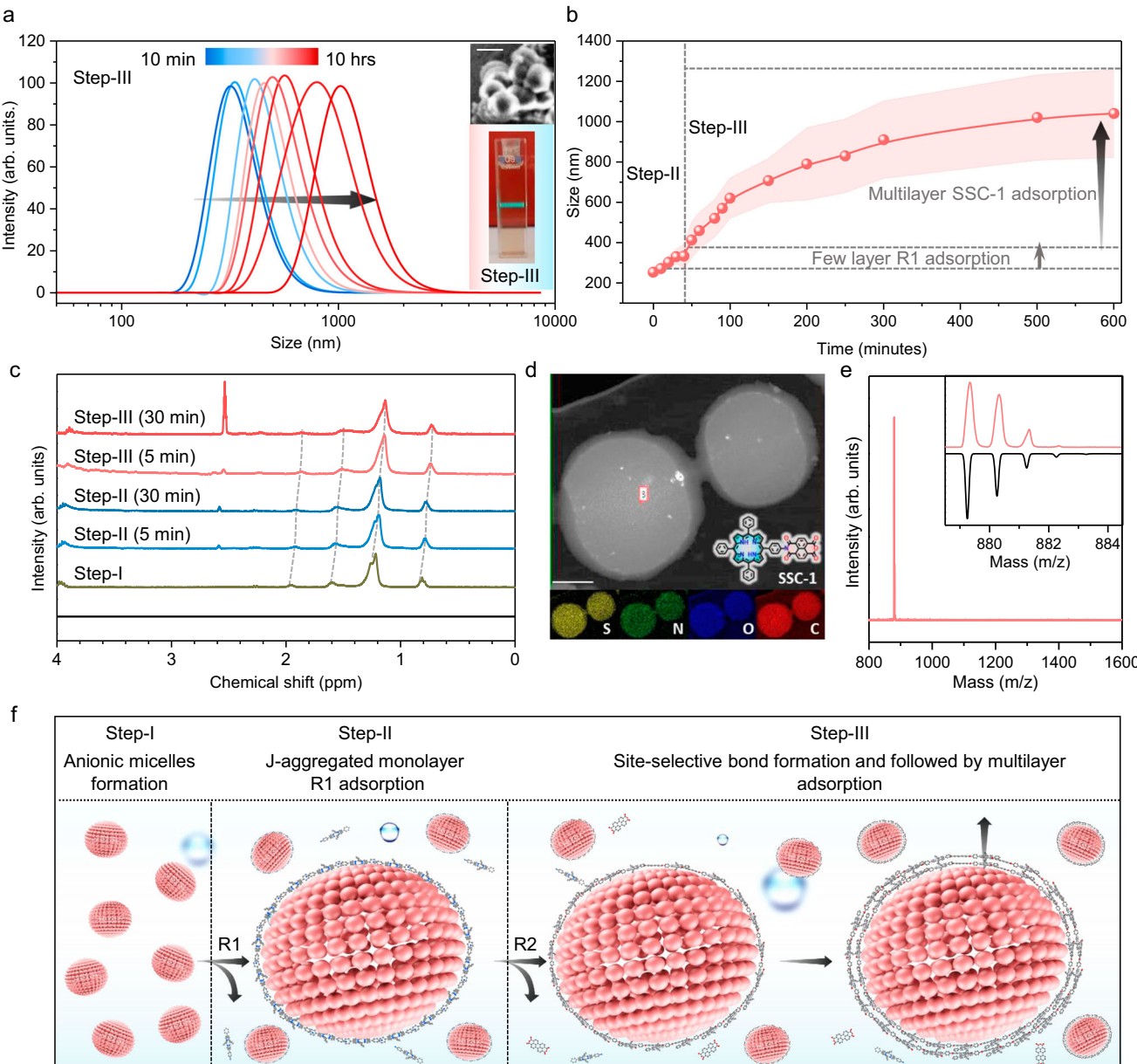

**Fig. 3 | Charge-driven micellar interface promoting multilayer adsorption.**
**a** Number-average size of the DLS-measured assembled structure of SOS-R1 micelles (Step-II) and following the addition of R3 (Step-III). Inset: Photographs of Step-III showcasing the Tyndall effect, with FE-SEM images illustrating the spherical morphology of the SOS-R1-R3 assembled structure (before washing with THF) with a scale bar of 1 μm. **b**, **c** In-situ DLS and ¹H NMR (300 MHz) spectroscopy analyses in D₂O showed across all three steps, from Step-I to Step-III. **d** Elemental mapping of SSC-1 on the SOS micelle surface using energy-dispersive X-ray spectroscopy (EDS) in scanning transmission electron microscopy (STEM), with a scale bar of 0.5 μm. **e** HR-MALDI-TOF mass spectra of the site-selective resultant compound SSC-1. **f** A schematic illustration of the sequential assembly for the multilayer growth of the site-selective product.

## Results

### Micellar interface mediated on-water surface chemical reactions

Designing an interface that mimics on-water surface chemistry is achieved through the self-assembly of charged surfactant molecules beyond their critical micelle concentration (CMC), forming micellar supramolecular structures and creating an environment with a hydrophobic core and surrounding water molecules at the surfactant head groups, which together form a stable interface for chemical reactions (Fig. 1a). The chemical reactivity and selectivity of the designed on-water surface utilizing these micelles were examined through a model reaction, as illustrated in Fig. 1b, c, employing a three-step sequential assembly process in an aqueous medium under ambient conditions at room temperature (Fig. 1c). The reaction progression from Step-I to Step-III was meticulously analyzed using in-situ techniques such as UV–Vis spectroscopy, dynamic light scattering (DLS), and nuclear magnetic resonance (NMR) spectroscopy, with an imide reaction serving as the representative example. Initially, in Step-I, micelles of sodium oleyl sulfate (SOS), an anionic surfactant, were constructed by dissolving SOS in an aqueous solution above its critical micelle concentration (CMC) of 1.7 mmol/L (Supplementary Fig. 1). Upon surpassing this concentration, the surfactant molecules spontaneously aggregate into micelles with an average hydrodynamic diameter of approximately 223 nm, as determined by DLS analysis (Fig. 2a). These micellar structures exhibited the Tyndall effect, indicating light scattering in the presence of micelles. Time-dependent DLS measurements showed that the hydrodynamic diameter of the

micelles remained constant over 30 min, confirming the stability of the micelle structure, as further supported by time-dependent surface tension measurements (Supplementary Fig. 2).

Following the formation of micelles, Step-II entails introducing the protonated 4-(5,10,15-triphenylporphyrin-20-yl)aniline (R1) molecule (at pH ~ 1.2) into the aqueous solution. The positive charge on the protonated R1 reactant molecules facilitates their electrostatic interaction with the negatively charged sulfate ($SO_{4-}$) head groups of the micelles (Supplementary Fig. 3). This interaction results in the pre-assembly of R1 molecules on the micelle surfaces. After the addition of R1 molecules for 30 min, this pre-assembled structure on the micelle surfaces leads to an increase in the micelle size to an average hydrodynamic diameter of ~295 nm, a change confirmed by DLS measurements, indicating that protonated R1 molecules effectively screen the negative surface charge of SOS anionic micelles through electrostatic interactions. (Fig. 2a). Field emission scanning electron microscopy (FE-SEM) reveals that the micelles retain their spherical structure with an average size of 300 nm and exhibit a strong Tyndall effect (Fig. 2a).

## Pre-assembly of R1 with distinct confined geometry on the micelle surface

Step-II is crucial as it facilitates the pre-organization of reactant molecules on the micelle surface. We explored Step-II by employing various in-situ (UV–vis, DLS spectroscopy) and ex-situ (FE-SEM, powder X-ray diffraction) characterization techniques. The UV–visible absorption spectra for R1 on the micelle surface showed a notable red shift in the Soret and Q-bands (466 and 688 nm, respectively), compared to those in the aqueous bulk, which peaked at 433 and 640 nm, respectively (Fig. 2b). In the attenuated total reflectance Fourier transform infrared (ATR-FTIR) spectroscopy analysis, a significant red shift was observed in the $NH_2$-wagging mode, shifting from 803 to 795 $cm^{-1}$ on the micelle surface relative to the aqueous bulk (Supplementary Fig. 4). These findings suggest the formation of a J-aggregated structure with a short slip distance in the R1 assembly, indicative of strong polarized-π interactions within the R1 assembled structure on the micelle surface[41–43]. In-situ UV–vis spectroscopy studies revealed that the R1 molecules fully assembled into the J-aggregated form within 30 min (Fig. 2c). One hour after the addition of R1 to the water subphase, the J-aggregated R1 assembly on the micelle was washed with cold tetrahydrofuran (THF, −20 °C) solvent and subsequently dried overnight for structural analysis. Powder X-ray diffraction (PXRD) analysis showed distinct diffraction peaks, indicating unit cell parameters of $a = 13.46$ Å, $b = 10.20$ Å, and $\gamma = 102.20°$ (Fig. 2d). The unit cell of the pre-assembled J-aggregate on the micelle surface is characterized by a significantly reduced lattice structure compared to the sizes of the individual R1 molecules, with the $a$ and $b$ parameters measuring 16.6 and 14.8 Å, respectively (Supplementary Fig. 5 and Supplementary Note 1). As a control experiment, we performed UV–visible spectroscopy and PXRD experiments on R1 both above and below the CMC of SOS surfactant, demonstrating that J-aggregate formation occurs only above the CMC, where micelles are present (Supplementary Figs. 5, 6 and Supplementary Notes 1, 2). This observation suggests a confinement of R1 molecules within a J-aggregated structure, facilitated by the micelle surface head group.

## Dynamic micellar interface induced by surface charge to favor multilayer adsorption

Next, we proceeded to Step-III, wherein the naphthalene tetra-carboxylic dianhydride (R2) reactant molecules were introduced into the aqueous solution. The addition of R2 induced a rapid and noticeable color transition from light green to dark orange within a minute, suggesting the efficient chemical reaction between R1 and R2 to form an imide bond. DLS analysis demonstrated that the overall size of the micelles increased, and after 10 h, the size reached ~1 μm (Fig. 3a–c). After completing Step-III, fluorescence confocal microscopy

measurements showed bright fluorescence from SSC-1 at the micelle boundaries (i.e., from the surface), confirming the hollow core (see Supplementary Fig. 7 and Supplementary Note 3 for detailed discussion). Elemental mapping using energy-dispersive X-ray spectroscopy (EDS) in scanning transmission electron microscopy (STEM) confirmed the presence of characteristic elements from SSC-1 and SOS micelles, supporting the uniform multilayer growth of SSC-1 on the micelle surface (Fig. 3d and Supplementary Fig. 8). FE-SEM revealed a spherical morphology with an average diameter of ~1 μm and exhibited a significant Tyndall effect (Supplementary Fig. 9 and Fig. 3a). Ten hours after the addition of R2 into the aqueous solution, the micellar assembly structure was washed with THF solvent to remove the surfactant. The product was then dried under a high vacuum, resulting in an isolated mass of 1.3 mg, which was subsequently used for further characterization. Atomic force microscopy (AFM) analysis confirmed the presence of circular sheet structures with a thickness of ~18 nm (Supplementary Fig. 9). The resultant compound was collected for analysis using matrix-assisted laser desorption/ionization-time-of-flight mass spectrometry (MALDI-TOF MS). A peak at an $m/z$ value of 879.28 was observed, showing an isotopic distribution corresponding to the imide site-selective product SSC-1, rather than the commonly expected two-sided imide product (Fig. 3e). This site-selective product, SSC-1, was further characterized using [1]H NMR, FTIR, and Raman spectroscopy, demonstrating exceptional selectivity (99%) and excellent yield (96%) (Supplementary Fig. 10). In contrast, when the same imide reactions were conducted in an aqueous solution under similar experimental conditions (pH, temperature, time, and concentration) without the use of SOS surfactant micelles, only the reagent molecules were identified by mass spectrometry (Supplementary Fig. 11). This control experiment clearly suggests that the confinement of porphyrin molecules on the water surface, facilitated by the micelles head groups, is crucial for enhancing their chemical reactivity (Supplementary Fig. 12 and Supplementary Note 4).

To elucidate the self-assembly kinetics and surface charge dynamics, we performed in-situ DLS and [1]H NMR spectroscopy analyses throughout all three steps. The [1]H NMR spectra revealed the characteristic peak of the SOS surfactant within the micelles. The introduction of R1 molecules led to an increase in micelle size and a downfield [1]H NMR shift, indicating a rise in electron density at the micelle surface (Fig. 3c). After several layers of R1 electrostatic adsorption, no further adsorption was observed after 30 min. This finding suggests that the protonated R1 molecules effectively screened the negative surface charge of the SOS micelles. Moreover, in Step-III, the addition of R2 triggered rapid growth in micelle size, which stabilized at 1080 nm after 10 h, indicating multilayer adsorption (Fig. 3b). The chemical reaction between R1 and R2 on the micelle surface resulted in the formation of an imide bond, releasing $H_3O^+$ ions and thus rendering the micelle surface negatively charged. This resulting negative charge on the micelles, along with the J-aggregated structure, promoted further adsorption of R1. The continuous formation of imide bonds and the associated release of charges facilitated the layer-by-layer growth of a multilayer product (Fig. 3f), as confirmed by a downfield [1]H NMR shift (for a detailed discussion, see Supplementary Fig. 13 and Supplementary Note 5). The observed surface charge behavior is consistent with previous studies on the SOS surfactant monolayer on the water surface[39].

## Exploring the generality of reactivity and site-selectivity on micelle surfaces

To investigate the generality of reactivity and site-selectivity on micelle surfaces as mimics of on-water surfaces, we extended this approach to a variety of chemical reactions involving porphyrin-based molecules. These reactions included the formation of imides, imines, amides, azoles (specifically thiazole, imidazole, and oxazole), pyridinium, and anhydrides. The stepwise sequential assembly

**Fig. 4 | Reactivity and selectivity in various chemical reactions on the micelle surface.** Overview of a wide range of different site-selective chemical reactions, including imide, imine, amide, azole, pyridinium, and anhydride reactions, encompassing a total of 14 different reversible and irreversible reactions, from SSR-1 to SSR-14.

reactions on the micelle surface were controlled by pH values. Depending on the required pH conditions for each reaction, we could choose between anionic or cationic micelles, both of which exhibited reactivity and selectivity. It is important to note that chemical reactions involving azoles (thiazole, imidazole, and oxazole) and imides typically require high temperatures (>170 °C) in bulk organic reactions[44]. By contrast, using our approach, we successfully demonstrated these reactions at room temperature under ambient conditions on anionic or cationic micelle surfaces. This extension encompassed 14 different site-selective reaction scopes, achieving exceptionally high selectivity (≥99%) and excellent yields in all site-selective reactions (for details, see Fig. 4 and Supplementary Figs. 14–34). Similarly, in the bulk aqueous synthesis, we achieved isolated yields of 93% (6.5 mg) for SSC-1 and 94% (5.4 mg) for SSC-10 using SOS and CTAB surfactant micelles, respectively (Supplementary Figs. 35–37 and Supplementary Note 6).

## Extension to 2D polymerization on the micelle surface

Following observations of reactivity and sequential assembly on the micelle surface in our model reactions, we extended this approach to 2D polymerization by using $(A_4 + B_2)$-type monomers, enabling the robust synthesis of crystalline 2D polymer sheets within micellar supramolecular structures through both reversible and irreversible reactions. To demonstrate the versatility of this approach, both anionic and cationic micelles were employed in the synthesis of 2D polyimine (2DPI) and 2D polybenzimidazole (2DPBI), respectively (Fig. 5a). Specifically, 2DPI was synthesized using 5,10,15,20-(tetra-4-aminophenyl)porphyrin (M1) and 2,5-dihydroxyterephthalaldehyde (M2) as monomers. For the synthesis of 2DPBI, 5,10,15,20-(tetra-4-carboxyphenyl)porphyrin (M3) and 1,2,4,5-benzenetetramine (M4) were used as monomers. 2D polymerization followed a similar step-wise reaction from Step-I to Step-III (for a detailed discussion, see Supplementary Fig. 38). After 18 h of 2D polymerization, the micellar assembly

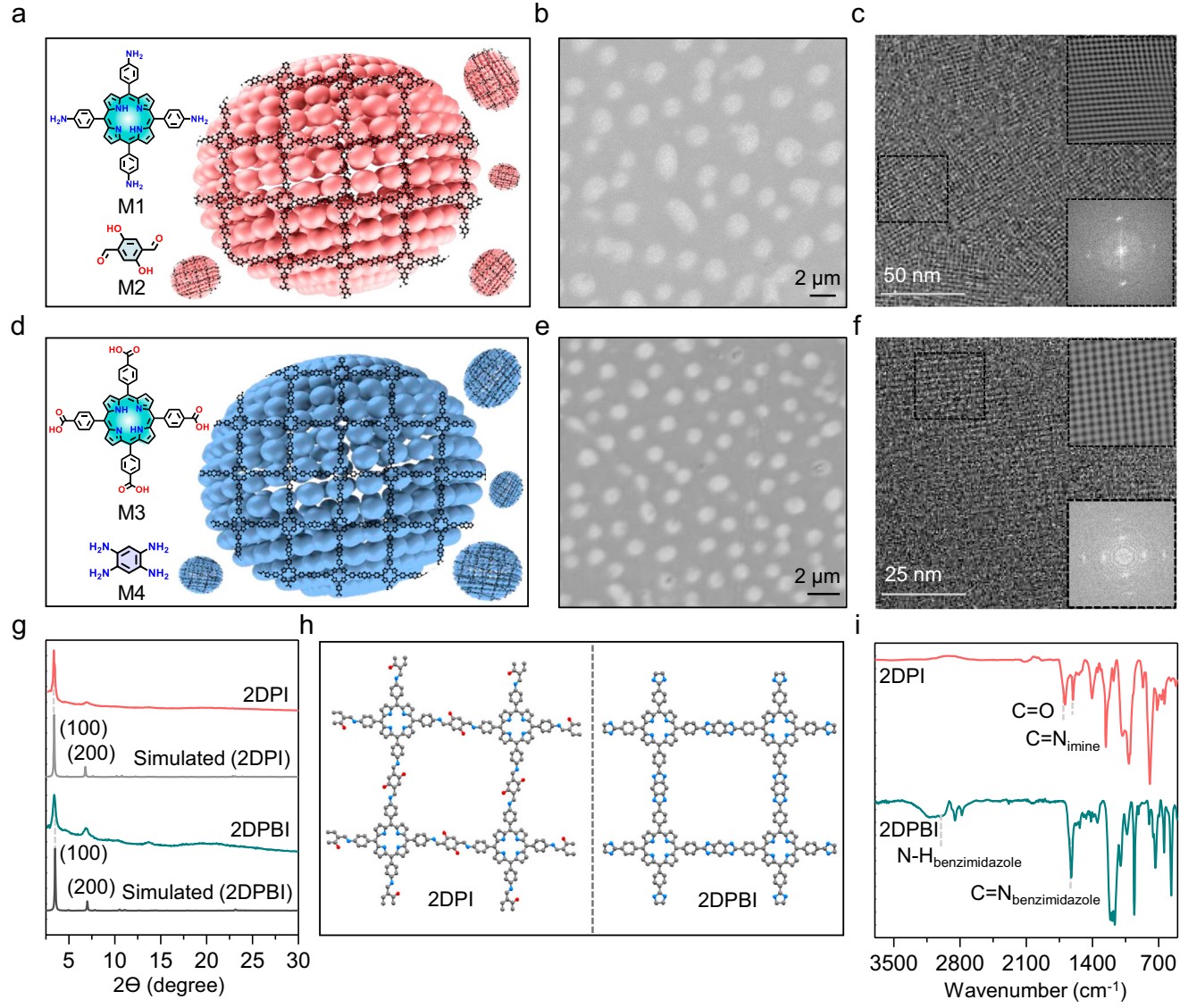

**Fig. 5 | Morphological and structural characterizations of synthesized 2DPs.**
**a** Schematic representation of the synthesis of 2DPI on anionic micelles. **b** Optical microscope images of 2DPI showing circular, sheet-like morphologies with uniform size distributions. **c** High-resolution transmission electron microscopy (HR-TEM) image of 2DPI, revealing a highly ordered square lattice. Inset: Corresponding fast Fourier transform (FFT) image and inverse fast Fourier transform (IFFT) pattern extracted from the HR-TEM image. **d** Schematic representation of the synthesis of 2DPBI on cationic micelles. **e** Optical microscope images of 2DPBI displaying circular, sheet-like morphologies with uniform size distributions. **f** HR-TEM image of 2DPBI revealing a highly ordered square lattice. Inset: Corresponding FFT image and IFFT pattern extracted from the HR-TEM image. **g** Comparison of X-ray diffraction (XRD) patterns for 2DPI and 2DPBI with simulated profiles. **h** Simulated molecular structure visualizations of 2DPI and 2DPBI. **i** Fourier-transform infrared (FTIR) spectral analysis highlighting the distinct chemical signatures of 2DPI and 2DPBI, with peaks labeled for functional groups relevant to each polymer.

structures were washed with THF solvent to remove the surfactant and any excess unreacted monomers, followed by drying at 100 °C overnight for further morphological and structural characterizations (Fig. 5b–i). As illustrated in Fig. 5b, e, the optical microscope images and the field-emission scanning electron microscopy (FE-SEM) images of 2DPI and 2DPBI exhibit circular, sheet-like morphologies with uniform sizes of ~2 and ~1 μm, respectively (Supplementary Fig. 39). Atomic Force Microscopy (AFM) measurements indicated a thickness of approximately 15.7 nm for 2DPI and ᴖ8.8 nm for 2DPBI, suggesting the formation of layer-stacked 2D polymers (Supplementary Fig. 40).

Further comparative analysis using Fourier-transform infrared (FT-IR) spectroscopy on the 2DPI showed the appearance of the −C=N bond at approximately 1618 cm$^{-1}$ and the imide C=O bond at around 1660 cm$^{-1}$. Additionally, the complete vanishing of the N–H stretch at

approximately 3315 cm$^{-1}$ was observed from M1. Similarly, for the 2DPBI, a broad peak at 3325 cm$^{-1}$ was observed, indicative of free N–H groups characteristic of hydrogen-bonded N–H within the 1,3-diazole (imidazole) ring. Simultaneously, pronounced new bands emerged at 1628 cm$^{-1}$, attributable to the C=N vibrations in the 1,3-diazole (imidazole) ring structure. The diminished intensity of the C=O band at 1697 cm$^{-1}$ suggests the successful formation of 1,3-diazole (imidazole) linkages (Fig. 5i). Solid-state $^{13}$C NMR spectroscopy further confirmed the presence of imine and imidazole linkages, verifying the linkage specificity of 2DPI and 2DPBI (Supplementary Fig. 41 and Supplementary Note 7). The crystallinity and lattice structure of the 2D polymers were further investigated through high-resolution transmission electron microscopy (HR-TEM) and powder X-ray diffraction (PXRD). HR-TEM images reveal a highly ordered square lattice with

lattice parameters of 25.42 Å for 2DPI and 25.56 Å for 2DPBI (Supplementary Fig. 42 and Supplementary Note 8). PXRD analysis showed distinct diffraction peaks, indicating unit cell parameters of $a$, $b = 25.56$ Å, and $\gamma = 90.0°$ for 2DPI, and $a$, $b = 25.28$ Å, and $\gamma = 90.0°$ for 2DPBI, demonstrating excellent agreement between the experimental results and the simulated structures (Fig. 5c–g).

In the sequential assembly process on micellar surfaces, the size of micelles plays a crucial role in achieving highly crystalline, large-area circular sheets and few-layer stacked 2D polymers. Larger micelles, such as those formed by the M1-SOS anionic surfactant (~318 nm), facilitate the pre-assembly of monomers over more extensive domain areas, thus promoting higher crystallinity in 2DPI. In contrast, smaller micelles, such as those formed by the M3-CTAB cationic surfactant (~106 nm), constrained by their limited surface area and increased angular curvature, typically result in reduced crystallinity and a tendency towards polycrystallinity with smaller circular-sheet domains in 2DPBI. This substantial difference in micelle size influences the morphological integrity and uniformity of the resulting 2D polymer sheets (Supplementary Fig. 42 and Supplementary Note 8). Consequently, we observed better crystallinity of 2DPI in the M1-SOS case, with circular-sheet domain sizes reaching ~2.0 μm, compared to the 1.0 μm domain size of 2DPBI in the M3-CTAB case.

It should be noted that comparing this mimicked on-water surface approach to traditional organic solution-phase synthesis, our method successfully synthesized 2DPI and 2DPBI at room temperature under ambient conditions in an aqueous solution (Supplementary Figs. 42–44 and Supplementary Notes 8–10). The achieved 2D polymer thin layers demonstrated high crystallinity and uniform thickness distribution (~15.2 nm for 2DPI and ~18.8 nm for 2DPBI) across all circular sheets. In contrast, traditional organic solution-phase synthesis requires volatile organic solvents, high temperatures (>100 °C for 2DPI and >170 °C for imidazole-linked amorphous polymer), and an inert atmosphere[40]. These stringent reaction conditions, combined with rapid aggregation and precipitation, often result in a lack of control over the thickness of synthesized 2DPs and their layer-stacked 2D COFs, ultimately leading to lower crystallinity with crystallite sizes smaller than 200 nm.

## Discussion

In summary, we have presented a strategy of mimicking on-water surface synthesis using self-assembled micellar structures formed from charged surfactant molecules, creating an environment where the hydrophobic core and surrounding water layer form an interface that facilitates chemical reactions akin to those observed in on-water surface chemistry. Employing such an on-water surface of micelles, we demonstrate reactivity and site-selectivity in 14 different reversible and irreversible chemical reactions. Extending this reactivity and the confinement of monomers with pre-organized structures on the micellar surface, our approach enables the aqueous synthesis of crystalline 2D polymers (2DPI and 2DPBI) thin layers under ambient conditions, yielding uniform circular-sheet size distributions on a micrometer scale and nanometer-scale thicknesses. From a perspective standpoint, this approach introduces a method for extending on-water surface chemistry to up-scalable aqueous synthesis, thereby broadening the accessibility of on-water surface chemistry for a wide range of chemical and materials syntheses.

## Methods

### General characterization

[1]H NMR spectra were recorded at room temperature with a Bruker Avance III HD 300 spectrometer operating at 300 MHz. DMSO-$d_6$ (DMSO, dimethylsulfoxide) was used as the solvent. [13]C CP solid-state NMR experiments were performed on a Bruker Avance NMR spectrometer operating at 75 MHz using a commercial 4 mm MAS NMR probe. The high-resolution matrix-assisted laser desorption/ ionization time-of-flight (MALDI-TOF) mass spectrometry was performed on a Bruker Autoflex Speed MALDI-TOF MS using trans-2-[3-(4-tert-butylphenyl)-2-methyl-2-propenylidene] malononitrile (DCTB) or dithranol as matrix. Optical microscopy (Zeiss), AFM (Bruker Multimode 8 HR), and TEM (Zeiss, Libra 200 KV) were used to investigate the morphology and structure of the samples. Thin films were deposited on a Si substrate for SEM measurements and on copper grids for TEM measurements. All the optical microscopy and AFM images were recorded on a 300 nm SiO$_2$/Si substrate. UV–Vis absorption spectra were recorded on a UV–Vis–NIR spectrophotometer Cary 5000 device at room temperature on a quartz glass substrate. Photoluminescence spectra were measured on the PerkinElmer fluorescence spectrometer LS 55. ATR-FTIR was performed on a Tensor II system (Bruker) with an attenuated total reflection unit, and the samples were prepared by depositing the thin films onto a copper foil. The trough was equipped with a platinum Wilhelmy plate, a Teflon dipper, and a pair of Delrin barriers. The hydrodynamic diameter and its distribution by intensity for the micelles in the aqueous colloidal dispersion were measured using the dynamic light scattering analyzer Zetasizer Nano-S90. The measurements were performed at ambient temperature and repeated three times per sample to ensure reproducibility and to provide a statistical evaluation of the studied systems. The fluorescence on the micelle's surface was studied using a Leica DMi8 inverted fluorescence microscope from Leica Microsystems CMS GmbH. Green laser with a wavelength of 552 nm was used for the excitation of the amphiphilies' surfaces and a fluorescence filter cube with a phase contrast filter was used to enhance the signal.

### Site-selective reaction SSR-1 on anionic micelle surface

Micelles of sodium oleyl sulfate (SOS), an anionic surfactant, were prepared by dissolving 20 mg of SOS (2.87 mmol/L) in an aqueous solution above its critical micelle concentration (CMC) of 1.7 mmol/L. This solution was placed in a 100 mL Duran glass bottle with a GL-45 cap. Then, R1 (1.59 μmol), dissolved in 20 mL of 0.15 M HCl aqueous solution, was injected into the bottle. After 45 min, R2 (7.94 μmol), dissolved in 20 mL of 0.05 M LiOH aqueous solution, was added to the bottle. After 10 h, the precipitate was washed sequentially with water and THF and then dried under a high vacuum. This resulted in the isolation of the pure site-selective product SSC-1, 1.3 mg, with a 96% yield.

## Data availability

The data that support the plots within this paper and other findings of this study are available from the corresponding author upon request.

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

## Acknowledgements

This work was financially supported by the EU Graphene Flagship (GrapheneCore3, no. 881603), ERC Consolidator Grant (T2DCP), H2020-MSCA-ITN (ULTIMATE, no. 813036), H2020-FETOPEN (PROGENY, 899205), CRC 1415 (Chemistry of Synthetic Two-Dimensional Materials, no. 417590517), SPP 2244 (2DMP), GRK2861 (no. 491865171), as well as the German Science Council and Center of Advancing Electronics Dresden. A.P. thanks Dr. Mike Hambsch for the fruitful discussion. The authors acknowledge the Center of Advancing Electronics Dresden and the Dresden Center for Nanoanalysis at TUD.

## Author contributions

X.F. and A.P. conceived and designed the project. A.P. planned the experimental sessions, synthesized, and characterized all site-selective compounds and 2DPs. S.G. and N.C. contributed to the in-situ and ex-situ NMR experiments. H.Y. performed FE-SEM and AFM analysis. A.M. performed TEM imaging. N.N.N., A.P., and M.A.A. performed DLS analysis. S.P. and E.B. performed ¹³C NMR analysis. M.A.A., A.H., and G.C. performed and analyzed the fluorescence microscopy. A.P. and X.F. co-wrote the manuscript with contributions from all the authors. All the authors discussed the results and commented on the manuscript.

## Funding

## Competing interests

The authors declare no competing interests.
