## [Transparent Peer Review file · Nature Communications]

Mimicking On-Water Surface Synthesis through Micellar Interfaces

Corresponding Author: Professor Xinliang Feng

Version 0:

Reviewer comments:

Reviewer #1

(Remarks to the Author)

The manuscript by Prason and co-workers report on using micellar interfaces for synthesis of porphyrin derivatives and their two-dimensional polymers. The authors show that porphyrin molecules can be linked with a variety of linkers and a number of different covalent linkages can be formed between porphyrins and reactants. The results are noteworthy and are potentially interesting to a broad chemical audience, ranging from on-water synthesis to two-dimensional polymers and frameworks. While I believe the results presented herein and the manuscript should be of interested to the readership of Nature Communications, I would suggest the authors consider the following points (which could overcome some potential flaws in the analysis and methodology).

First of all, my understanding of the manuscript is that is presenting a new synthetic methodology. From this point of view, there are some suggestions the authors should consider:

- what is the scale on which the reactions can be performed? The authors report synthesis on mg scale, while for the method to be useful, it should be discussed what is the max scale the reaction (of discrete molecules and polymers) can be performed? Does scaling up have impact on selectivity and on the reaction yields?
- it seems that the reaction scope is limited only on porphyrins (see also point later regarding mechanism), which deserves some discussion. Is this true and, if yes, what is the reason? Can other chromophore be used to prepare discrete molecules and polymers with the methodology presented in the manuscript?
- given the new methodology, the authors should consider discussing it in comparison with previous methods. Especially, compared to LB and SMAIS methodologies, what are the benefits and what are the limitations?

Second, regarding the mechanism:

- the protonation state of the porphyrin seems to be playing a role of utmost importance, with pH being critical. The green color of the initial porphyrin solution (at pH 1.2) is indicative of protonated pyrrole of the porphyrin ring (pKa around 2-3). Therefore the change in color seems to be connected to change in protonation state, which then leads to aggregation. Additionally, the spectra presented in Figure 2c are misleading, at least to me: They are presented as kinetics and should be plotted accordingly as there seems to be sharp transitions that should be discussed. The grey traces seem to be a gradual change from the grey traces to the green one, and it unclear which traces belong to which photograph in Figure 2c.
- the discussion about NMR is not convincing and it is unclear how layer-by-layer growth is visualized by NMR. There is no detailed discussion in Supplementary Figure 12 as mentioned in the main text.
- regarding the mechanism, have the authors considered performing confocal microscopy to visualize the assemblies and their location on micelles?
- the discussion micelle size and formation of extended 2D polymers is not convincing. To me, it looks more likely that the change of surfactant charge is affecting the size of 2D polymers, rather than micelle size. Further discussion of interaction of porphyrins with cationic and anionic surfactants, as well as linking micelle size with 2D polymers is encouraged.

Finally, regarding some experimental details:

- some experiments, such as DLS (and others), should be reported as average of various measurements and error should be visible on the graphs.
- how was the selectivity calculated? It is unclear what the expected byproducts are
- was the yield calculated based on the mass measured on the powders that were used for recording NMRs reported in the ESI? In this case, it is very likely that the yields are overestimated, since the NMR spectra have clearly some impurities.

Reviewer #2

(Remarks to the Author)

The authors presented a novel method to mimic on-water surface chemistry by micelles. Using the characteristics of charged surfactant molecules, a micelle structure simulating the air-water interface was formed to achieve unique reactivity and high selectivity conversion of 14 reactions. The method was extended to 2D polymerization, and the water synthesis of crystalline 2D polymer thin layer was realized. This strategy is novel and meaningful, broadening the accessibility of on-water surface chemistry in a wide range of chemical and material synthesis. However, some revision to following issues should be provided for consideration before publication.

1. In the fifth paragraph of result and discussion, "The continuous formation of imide bonds and the associated release of charges facilitated the layer-by-layer growth of a multilayer product (Fig.3f), as confirmed by a downfield ^1H NMR shift (for a detailed discussion, see Supplementary Fig. 12)." Please verify that the relevant pictures mentioned are correct.
2. The time-dependent surface tension measurements (Supplementary Fig. 2) can only indicate that the micelles still exist, but cannot explain the high stability of the micelles. And there is no clear concentration of SOS. I think there is not rigorous expression here.
3. The micelle surface head group promotes the formation of R1 molecules within a J-aggregated structure. It would be better to add the case with or without micelles in Fig. 2d as a comparison.
4. In the part of extension to 2D polymerization on the micelle surface, the author mentions the (A4+B2)-type monomers, and its related information is not found in the manuscript, which I think should be added.
5. In the sequential assembly process on micellar surfaces, the size of micelles plays a crucial role in achieving highly crystalline, large-area circular sheets and few-layer stacked 2D polymers. However, there is no discussion on the effect of adjusting micelles size on the formation of 2D polymers in the current characterization, which needs to be clarified in the revision.
6. Why choose SOS and CTAB instead of other charged surfactants? Can other charged surfactants form similar 2D polymer sheets?
7. Some expressions need to be added with references, such as the sixth paragraph of result and discussion, "imides typically require high temperatures ($>170^\circ\text{C}$) in bulk organic reactions".
8. Adding statistics on the size of 2DPI and 2DPBI in Figures 4b and 4e will be more complete.

Reviewer #3

(Remarks to the Author)

In this manuscript, A. Prasoon et al. presented a new approach that mimics on-water surface chemistry using micelles. The steps of this approach include the formation of charged surfactant molecules beyond their CMC, assembly of one type of reaction molecule, and chemical reactions taking place on the surfaces of micelles. This approach created an environment where the hydrophobic core and surrounding water layer form an interface that facilitates chemical reactions akin to those observed in on-water surface chemistry. However, there are some scientific and technical issues that prevent this reviewer from recommending acceptance of the manuscript for publication.

- (1) The amino-substituted porphyrin is a 2D molecule. On the surface of the micelles, the molecules will be parallel or perpendicular to the surfaces of micelles?
- (2) The authors claimed that the 2DPI and 2DPBI were highly crystalline structured. What about the polymerization degree? Crystallinity of the polymers is highly related to their polymerization degrees and should be compared with that of the materials obtained under a regular way.
- (3) About Figure 1a, the left and middle images of Figure 1a have nothing to do with this research. Figure 1 should be simplified. If any reactions or procedures are not related to this research, they should be removed.
- (4) The inset of Figure 2a shows an SEM image, which is not clear and lacks a scale bar. TEM should give clearer images. The images obtained before and after porphyrin assembly should be compared.
- (5) The authors used UV-visible spectroscopy to confirm the formation of J-aggregates. The absorption of porphyrin is very sensitive to the solution circumstance. An absorption spectrum of porphyrin should be obtained at a SDS concentration lower than the CMC.
- (6) THF was used to wash the porphyrin J-aggregates. What about the solubility of the porphyrin in THF?
- (7) XRD data were used to confirm the formation of porphyrin J-aggregates (1.346 and 1.020 nm vs. 1.66 and 1.48 nm). How were the values of 1.66 and 1.48 nm obtained? Measurement should be carried out with control samples, which was obtained at a SDS concentration lower than the CMC.
- (8) The color change may not due to the reaction. Porphyrin shows different color at alkaline or acid media. A control test should be performed with only R2.
- (9) On page 6, "FE-SEM revealed a spherical morphology with an average diameter of $\sim 1\ \mu\text{m}$ and a thickness of $\sim 18\ \text{nm}$ ". It is hard to draw such conclusion.
- (10) The SEM image in Figure 3a is not clear and lacks a scale bar.
- (11) Line 195 on page 8, "which stabilized at 1020 nm after 5 hours". However, the data in Figure 3a do not show 1020 nm.
- (12) No substantial evidences supported the formation of a hollow structure with a hydrophobic core surrounded by water. The authors are advised to conduct further investigations into this structure.
- (13) Line 197 on page 8, "releasing H_3O^+ ions". R2 is an anhydride, which is sensitive to water. Why did it not react with water when it added into the reaction mixture? Suppose R2 does not react with water, it is H_2O that is released after the reaction between R1 and R2. Where is the proton from? R2 has two functional groups, why did the other functional group of R2 not react with another R1?
- (14) In the model reaction, only one functional group of R2 reacted with R1. Why could the polymerization take place?
- (15) Figure 4b and 4e only show particles, lacking direct evidences for sheet-like morphologies of the products.
- (16) Scale bars should be given in the AFM images. They look like particles. Cross-section analysis is needed.

(17) From Line 264 to 270 on page 10, evidences for achieving the cell parameters are not enough. In Figure 4c, the angle between the different directions of lattices is not 90 degrees.

(18) The particles shown in Figure 4b and 4e were not uniform in size. Distribution analysis may be required.

(19) Can the authors obtain the number of layers in the stacked 2D polymers?

(20) In figure S32a, where is the material, dark area or light area?

Version 1:

Reviewer comments:

Reviewer #1

(Remarks to the Author)

The authors have answered my initial concerns and made the appropriate changes. I would suggest publication at this stage.

Reviewer #2

(Remarks to the Author)

The manuscript has been well revised and could be accept. Following the revisions, the discussion is more comprehensive, and the experimental approach is more robust than previously. The novelty of this research is significant, and it is expected to attract the interest of readers of Nature Communications. It is hoped that the author can successfully extend this strategy to other surfactants in the future to achieve more in-depth research.

Reviewer #3

(Remarks to the Author)

The authors have modified the manuscript following the comments raised in the first round of review. While most of the comments have been reasonably responded and been reflected in the revised manuscript, there are still a couple of concerns.

Comment 1: The authors explained how difficult to measure the polymerization degree of a 2D polymer, which is acceptable. What about the crystallinity of their material when compared with the material obtained under a regular way?

Comment 2: Figure 2d shows the PXRD data of the R1 assembled structure. The indexing may be wrong because (100) was used for indexing two different peaks. Please check them carefully.

Version 2:

Reviewer comments:

Reviewer #3

(Remarks to the Author)

I am satisfied with the revisions made by the authors. Therefore, I recommend acceptance of this version of the manuscript for publication.

Detailed response to the comments from the reviewers

Reviewer #1 (Remarks to the Author):

General comment: *The manuscript by Prasoon and co-workers report on using micellar interfaces for synthesis of porphyrin derivatives and their two-dimensional polymers. The authors show that porphyrin molecules can be linked with a variety of linkers and a number of different covalent linkages can be formed between porphyrins and reactants. The results are noteworthy and are potentially interesting to a broad chemical audience, ranging from on-water synthesis to two-dimensional polymers and frameworks. While I believe the results presented herein and the manuscript should be of interest to the readership of Nature Communications, I would suggest the authors consider the following points (which could overcome some potential flaws in the analysis and methodology).*

Response: We greatly appreciate the reviewer for the highly positive and insightful comments, which have been invaluable in the revision of our manuscript.

First of all, my understanding of the manuscript is that is presenting a new synthetic methodology. From this point of view, there are some suggestions the authors should consider:

Comment 1: *What is the scale on which the reactions can be performed? The authors report synthesis on mg scale, while for the method to be useful, it should be discussed what is the max scale the reaction (of discrete molecules and polymers) can be performed? Does scaling up have impact on selectivity and on the reaction yields?*

Response: We appreciate the reviewer's comment regarding the scalability of the reactions. In response to your suggestion, we performed scale-up reactions for the synthesis of site-selective compounds using both anionic and cationic surfactant micelles, demonstrating the generality of the on-water surface micelle-mediated reactions.

For the anionic SOS surfactant micelles, we scaled up the synthesis of SSC-1 in the typical 250 mL reaction flask, achieving an isolated yield of 93% (6.5 mg). Similarly, for the cationic CTAB surfactant micelles, we scaled up the synthesis of SSC-10, obtaining an isolated yield of 94% (5.4 mg). Here, we would like to mention that the solubility of the reactants, 4-(5,10,15-triphenylporphyrin-20-yl)aniline (R1) and 5-(4-carboxyphenyl)-10,15,20-(triphenyl)porphyrin (R3), is low in water, which limited the scale-up of the reactions and resulted in these product quantities. The selectivity in the scaled-up reactions (SSC-1 and SSC-10) was maintained at over 99%, as confirmed by MALDI-TOF MS and NMR spectroscopy, with no evidence of two-sided imide or imidazole products. Only the site-selective products, SSC-1 and SSC-10, were observed (Figures R1-R3).

Additionally, we scaled up 2D polymerization reactions (2DPI and 2DPBI) using both cationic and anionic surfactant micelles in the typical 250 mL reaction flask to further demonstrate the versatility of the approach. For 2D polymerization, we used porphyrin monomers 5,10,15,20-(tetra-4-aminophenyl)porphyrin (M1) and 5,10,15,20-(tetra-4-carboxyphenyl)porphyrin (M3), which have higher solubility in water due to their increased protonation sites. The isolated yields for the 2D polymers, 2DPI and 2DPBI, were 24.9 mg and 31.2 mg (92% and 94% yield), respectively. Solid-state ¹³C NMR spectroscopy confirmed the presence of imine and imidazole linkages, verifying the linkage specificity of the polymerized products (Figure R4). These results indicate that while solubility limits the scale of some reactions, the methodology maintains high selectivity and efficiency.

Figure R1 | A schematic illustrating the site-selective chemical reaction for scale-up of (a) SSC-1 and (b) SSC-10 on the water surface, including the expected two-sided product. (c) HR-MALDI-TOF mass spectra of the site-selective products SSC-1 and SSC-10.

Figure R2 | ^1H NMR spectra (300 MHz) of SSC-1 in DMSO-d_6 solvent.

Figure R3 | ^1H NMR spectra (300 MHz) of SSC-10 in DMSO-d_6 solvent.

2D polymerization on the micelle surface

Figure R4 | Schematic representations of the synthesis of 2DPI on anionic micelles and 2DPBI on cationic micelles, along with the solid-state ^{13}C NMR spectra of 2DPI and 2DPBI.

ACTION:

Supplementary Information: Figures R1-R3 have been incorporated into the revised SI as Supplementary Figs. 35-37, and Figures R4 as Supplementary Fig. 41.

Main Text: We have added the following sentence to the main text (page 9): “Similarly, in the bulk aqueous synthesis, we achieved isolated yields of 93% (6.5 mg) for SSC-1 and 94% (5.4 mg) for SSC-10 using SOS and CTAB surfactant micelles, respectively.”

Comment 2: *It seems that the reaction scope is limited only on porphyrins (see also point later regarding mechanism), which deserves some discussion. Is this true and, if yes, what is the reason? Can other chromophore be used to prepare discrete molecules and polymers with the methodology presented in the manuscript?*

Response: We appreciate the reviewer's comment regarding the reaction scope. The key to our approach lies in designing monomers that meet two fundamental requirements: (i) reactants with pH-dependent charges, and (ii) reagents with charges opposite to those of the micelles. The presence of charged components facilitates electrostatic interactions with oppositely charged surfactants, while J-aggregated confinement enhances the reactivity of the targeted site, enabling selective bond formation.

We selected porphyrins as a proof of concept because of their unique properties, such as pH-dependent charge modulation and J-aggregated confinement, which align well with the criteria required for our methodology. However, although our current study focused specifically on porphyrins, the principles underlying our approach can be extended to other chromophores or molecules with similar pH-responsive behavior and charge characteristics. For instance, we are currently working on different monomers with various symmetries, such as (A₃+B₂)-type monomers for hexagonal lattices, using larger micelles (over tens of microns) formed by mixed surfactants. This approach could create micelles with larger accessible surface areas, potentially facilitating the growth of various types of crystalline 2D polymers.

Figure R5 | Schematic representation of the synthesis of a 2D imidazole polymer with (A₃+B₂)-type monomers arranged in a hexagonal lattice within anionic mixed micelles.

Comment 3: *Given the new methodology, the authors should consider discussing it in comparison with previous methods. Especially, compared to LB and SMAIS methodologies, what are the benefits and what are the limitations?*

Response: We appreciate the reviewer's suggestion. Below, we discuss the benefits and limitations of our approach in comparison to these established methods:

Benefits of Our Methodology:

Despite significant advancements in on-water surface synthesis using LB and SMAIS, a critical challenge remains their limited accessibility, as these methods rely on a stable air-water interface. This limitation confines the unique reactivity and spatial geometry to two-dimensional spaces, which can hinder the development of broader synthetic applications.

Our current approach successfully mimics on-water surface chemistry and extends it to bulk aqueous synthesis, providing greater accessibility for reactions. By utilizing the on-water surface of micelles as the reaction medium, we overcome the spatial constraints inherent in LB and SMAIS methodologies, enabling bulk synthesis of site-selective compounds and 2D polymers.

Limitations of Our Methodology:

A potential limitation of our approach is that the 2D polymer crystal domain size is constrained by the surface area of the micelles. The circular curvature of micelles can reduce crystallinity, often resulting in a polycrystalline nature in the produced 2D polymers. In contrast, LB and SMAIS provide a stable, homogeneous, and flat water surface, which promotes the growth of more uniform and highly crystalline 2D polymer films.

We are currently working on extending the strategy of on-water surface chemistry by designing larger micelles (above tens of microns) with different shapes (ellipsoidal, cylindrical, and unilamellar vesicles) using mixed surfactants and Gemini surfactants. Such approach can allow to create micelles with larger accessible surface areas, potentially facilitating the growth of single-crystalline 2D polymers.

ACTION:

Supplementary Information: The related discussions are included in Supplementary Fig. 42.

Second, regarding the mechanism:

Comment 4: *The protonation state of the porphyrin seems to be playing a role of utmost importance, with pH being critical. The green color of the initial porphyrin solution (at pH 1.2) is indicative of protonated pyrrole of the porphyrin ring (pKa around 2-3). Therefore the change in color seems to be connected to change in protonation state, which then leads to aggregation. Additionally, the spectra presented in Figure 2c are misleading, at least to me: They are presented as kinetics and should be plotted accordingly as there seems to be sharp transitions that should be discussed. The grey traces seem to be bThere does not seem to be a gradual change from the grey traces to the green one, and it unclear which traces belong to which photograph in Figure 2c.*

Response: We appreciate the reviewer's observation regarding the critical role of protonation in the behavior of porphyrins and the influence of pH. The protonation state of the porphyrin significantly affects the electrostatic interactions between the porphyrin and the SO₄⁻ head group of the SOS surfactant. The protonation state directly influences the aggregation behavior, as changes in protonation alter electrostatic interactions, driving aggregation.

Regarding the spectra presented in Figure 2c, we acknowledge the reviewer's concerns about the representation of the data and the need for a clearer depiction of the kinetic process. In our initial experiments, we monitored the UV-Vis spectra in-situ by adding a higher volume of stock porphyrin solution (1 ml) and mixing rigorously in the cuvette. This approach led to sharp transitions due to the rapid electrostatic interactions between the porphyrin monomers and the abundant anionic micelles in the bulk solution, resulting in fast diffusion of porphyrin monomers over the micelle surfaces. These rapid interactions caused abrupt spectral changes.

To address this issue, we revised our experimental approach to allow for a more gradual observation of the spectral changes. In the new data presented in the manuscript, we performed UV-Vis measurements directly in a cuvette by gently adding a lower volume of stock porphyrin solution (0.5 ml) with minimal mixing. This adjustment allowed for more controlled and

gradual observation of the changes in the spectra, providing a clearer representation of the kinetic process. Please note that the transition is very fast, as shown in Figure R6, where the formation of J-aggregates occurs within three minutes (indicated by the blue dashed line). This revised approach better correlates the spectral changes with the corresponding photographs in Figure 2c.

Figure R6 | In-situ UV-Vis spectroscopy studies of Step-II, illustrating the short-slip distance in the J-aggregated R1 structure.

ACTION:

Main text: Figure R6 is now Figure 2c in the revised main text.

Comment 5: *The discussion about NMR is not convincing and it is unclear how layer-by-layer growth is visualized by NMR. There is no detailed discussion in Supplementary Figure 12 as mentioned in the main text.*

Response: We appreciate the reviewer's comment regarding the NMR discussion and the visualization of layer-by-layer growth. The reaction progression from Step-I to Step-III was thoroughly analyzed using in-situ techniques such as Dynamic Light Scattering (DLS) and Nuclear Magnetic Resonance (NMR) spectroscopy, with an imide reaction (SSC-1) serving as a representative example of the layer-by-layer assembly mechanism.

Step-I: In this step, anionic micelles of SOS were formed by dissolving SOS in an aqueous solution. The ^1H NMR spectra showed characteristic peaks corresponding to the SOS surfactant within the micelles, providing a reference for monitoring subsequent changes.

Step-II: Following the formation of micelles, protonated 4-(5,10,15-triphenylporphyrin-20-yl)aniline (R1) molecules were introduced into the aqueous solution at pH \sim 1.2. The ^1H NMR spectra showed a downfield shift in the characteristic peak of the SOS surfactant within 30 minutes of introducing R1, indicating increased electron density at the micelle surface due to the adsorption of the R1 molecules. This shift is attributed to the electrostatic interaction between the negatively charged head groups (SO_4^-) of the SOS surfactant and the protonated R1 molecules. Concurrently, time-dependent DLS measurements revealed an increase in micelle size upon the introduction of R1, confirming the adsorption process. After several layers of R1 were adsorbed, no further adsorption was observed after 30 minutes, as indicated by the stabilization of the hydrodynamic diameter of the R1-SOS micelle assembly (average size 325 nm) (Figure R7a). This stability suggests that the protonated R1 molecules effectively screened the negative surface charge of the SOS micelles, leading to charge neutralization.

Step-III: The addition of naphthalene tetracarboxylic dianhydride (R2) molecules triggered a rapid increase in micelle size, as observed in time-dependent DLS measurements, which stabilized at approximately 1080 nm after 10 hours (Figure R7a). This size increase is indicative of multilayer adsorption, driven by the chemical reaction between R1 and R2 on the micelle surface, resulting in imide bond formation and the release of H_3O^+ ions. The release of protons from the reaction sites rendered the micelle surface negatively charged, facilitating further adsorption of positively charged R1 molecules from the bulk aqueous solution.

Figure R7 | (a) Average hydrodynamic diameter size over time measured by in-situ time-dependent DLS. (b) Elemental mapping using energy dispersive X-ray spectroscopy (EDS) in scanning transmission electron microscopy (STEM).

The negative surface charge, along with the well-defined J-aggregate template structure on the micelles, served as a driving force for the continued adsorption of R1 (Figure R7). The ongoing formation of imide bonds and subsequent charge release perpetuated the layer-by-layer growth of the multilayer site-selective product (SSC-1), which reached equilibrium after 10 hours, with a hydrodynamic diameter of approximately 1080 nm, as shown by the time-dependent DLS analysis. The observed surface charge behavior is consistent with prior studies on SOS surfactant monolayers on the water surface, including findings from ultrafast phase-sensitive interface-selective nonlinear vibrational spectroscopy, particularly sum-frequency generation (SFG) spectroscopy (*Nat Commun* **14**, 8313 (2023)). Additionally, elemental mapping using Energy Dispersive X-ray Spectroscopy (EDS) in Scanning Transmission Electron Microscopy (STEM) confirmed the presence of characteristic elements from SSC-1 and SOS micelles, supporting the uniform multilayer growth of SSC-1 on the micelle surface (Figure R7b). The

EDS mapping also highlighted the hollow hydrophobic core of the micelles, further validating the layer-by-layer growth mechanism observed. Overall, the combined in situ ^1H NMR, DLS and EDS-STEM studies suggest a clear layer-by-layer growth process, as well as the dynamic changes in surface charge that drive this assembly mechanism.

Figure R8 | A schematic illustrating a site-selective chemical reaction on the surface of SOS micelles.

ACTION:

Supplementary Information: Figures R7-R8 have been incorporated into the modified SI as Supplementary Fig. 12 and 13.

Comment 6: *Regarding the mechanism, have the authors considered performing confocal microscopy to visualize the assemblies and their location on micelles?*

Response: We appreciate the reviewer's constructive suggestion. Following your suggestion, we employed confocal fluorescence microscopy to directly observe the fluorescence emitted from the SSC-1 on the micelle surface in an aqueous solution, thus assessing the presence of hollow micelles.

Upon completing Step-III, where the SSC-1 product was formed on the micelle surface, the hollow micelles exhibited bright fluorescence signals along their surface boundaries when viewed under the microscope. By exciting the sample at a specific wavelength (552 nm) corresponding to the excitation spectrum of SSC-1, the emitted light was captured and analyzed. Adjusting the focal plane of the microscope (as shown in Figure R9 from panels a to c) allowed us to follow and visualize the fluorescence pattern at the outer edge of the micelles. This real-time visualization of the micelle surface is further illustrated in a supplementary video.

Regions of irregular or weak fluorescence, appearing as dark areas, indicate zones not illuminated by the fluorescent laser. For example, in the micelle labeled with a red circle in panels a, b, and c, adjusting the focal plane towards the observer revealed high fluorescence intensity at the edges but not within the central black region, which gradually closed as the focal plane was moved further (panel b). The black area signifies the absence of emitted fluorescence, confirming that the internal core of the micelle is hollow and does not contain any reactant molecules (R1 or R2) within its hydrophobic core.

To provide a more comprehensive understanding, we recorded a video demonstrating the dynamic behavior of the formed micelles in real-time, showcasing the spatial distribution and structural integrity of the assemblies. Additionally, quantitative analysis of the fluorescence intensity was performed to estimate the density and distribution of the SSC-1 product on the micelle surface (Figure R9-d). This approach provided a clear and direct method to monitor

and verify the functionalization of hollow micelles with the SSC-1 product. Gray-scale visualization of the emitted fluorescence intensity upon laser excitation confirmed no light intensity within the hollow core of the micelles, indicating the absence of SSC-1 product inside the micelles.

Figure R9 | Sequences (a, b, c) of the emitted fluorescence at different focal levels. The black color indicates the hollow core of the micelles and the absence of the SSC-1 product inside. (d) Gray-scale visualization of the emitted fluorescence intensity upon laser excitation, showing no light intensity recorded in the hollow body of the micelles.

ACTION:

Supplementary Information: Figure R9 has been incorporated into the revised SI as Supplementary Fig. 7.

Main Text: We have added the following sentence to the main text (page 6): “After completing Step-III, fluorescence confocal microscopy measurements showed bright fluorescence from SSC-1 at the micelle boundaries (i.e., from the surface), confirming the hollow core (see Supplementary Fig. 7 for detailed discussion).”

Comment 7: *The discussion micelle size and formation of extended 2D polymers is not convincing. To me, it looks more likely that the change of surfactant charge is affecting the size of 2D polymers, rather than micelle size. Further discussion of interaction of porphyrins with cationic and anionic surfactants, as well as linking micelle size with 2D polymers is encouraged.*

Response: We appreciate the reviewer’s insights and concerns. We agree that the interaction between porphyrins and surfactants, along with the effect of surfactant charge, plays a crucial role in determining the overall size and structure of the 2D polymers. Below, we provide a more detailed explanation of these interactions and how micelle size and surface charge contribute to 2D polymer synthesis.

Influence of surface charge density and micelle size on 2D polymer synthesis:

1. **Surface charge density:** The surface charge density of micelles is a critical factor influencing the adsorption and assembly of porphyrin molecules on the micelle surface. The charge density follows a double-layer model (*Advances in Condensed Matter Physics* **2015**, *2015* (1), 151683), where the charged surfactant head groups interact with oppositely charged porphyrins, facilitating their adsorption onto the micelle surface. High surface charge density enhances the electrostatic interactions, promoting the formation of ordered assemblies and ultimately influencing the size and crystallinity of the resulting 2D polymers.

2. **Impact of micelle size:** The curvature of the micelle surface, particularly angular curvature, significantly influences the packing and arrangement of porphyrin molecules. Larger micelles

have lower curvature, providing a more planar surface, which allows for a more even distribution with less distortion, better organization of the porphyrin self-assembled structure, and stable adsorption of porphyrins. This structured assembly is essential for developing 2D polymers with larger domain sizes and higher crystallinity. In our study, larger micelles, such as those formed by the M1-SOS anionic surfactant (~318 nm), facilitate the pre-assembly of monomers over more extensive domain areas, promoting higher crystallinity in 2DPI. In contrast, smaller micelles, such as those formed by the M3-CTAB cationic surfactant (~106 nm), are constrained by their limited surface area and increased angular curvature, leading to reduced crystallinity and a tendency towards polycrystallinity with smaller circular-sheet domains in 2DPBI.

Key points on interaction of porphyrins with charged surfactants: The interaction between porphyrins and surfactants is highly dependent on charge compatibility and pH conditions, which modulate the surface charge of the micelles and influence the assembly process:

pH-dependent charge states: The protonation state of the porphyrin substituents dictates the charge on the porphyrins and therefore their affinity for the surfactant, affecting the strength of electrostatic interactions and the resulting aggregation behavior.

Amino-substituted porphyrin (R1) with anionic SOS surfactant: Under acidic conditions, R1 is positively charged and interacts strongly with the negatively charged SO_4^- groups of SOS micelles. This interaction, driven by strong electrostatic attraction and polarized- π interactions among the porphyrin rings, leads to the formation of J-aggregates.

Carboxy-substituted porphyrin (R3) with cationic CTAB surfactant: Under basic conditions, the carboxylic acid groups of R3 are deprotonated, making the porphyrin negatively charged. R3 then interacts electrostatically with the positively charged ammonium groups on the CTAB micelles. This strong electrostatic attraction leads to the adsorption of R3 on the micelle surface, where the planar nature of porphyrins facilitates close packing and strong intermolecular interactions, promoting the formation of J-aggregates.

Overall, both the size and surface charge density of micelles are fundamental factors in the formation of 2D polymers. While surface charge density dictates the strength and distribution of electrostatic interactions between the porphyrins and the surfactant head groups, the micelle size and curvature influence the spatial arrangement, stability, and crystallinity of the adsorbed porphyrins.

ACTION:

Supplementary Information: The discussion on the influence of surface charge density and micelle size on 2D polymer synthesis is included in Supplementary Fig. 42.

Finally, regarding some experimental details:

Comment 8: *Some experiments, such as DLS (and others), should be reported as average of various measurements and error should be visible on the graphs.*

- *how was the selectivity calculated? It is unclear what the expected byproducts are*
- *was the yield calculated based on the mass measured on the powders that were used for recording NMRs reported in the ESI? In this case, it is very likely that the yields are overestimated, since the NMR spectra have clearly some impurities.*

Response: We appreciate the reviewer's suggestion to present data, such as DLS measurements, as averages of multiple replicates and to include error bars on the graphs. In our revised manuscript, the DLS data and other relevant measurements are the mean of three independent measurements, with error bars representing the standard deviation.

Selectivity was evaluated using MALDI-TOF MS and NMR spectroscopy, where only the site-selective products were detected, with no evidence of bisubstituted (two-sided) products. For the anionic SOS surfactant micelles, we scaled up the synthesis of SSC-1, achieving an isolated yield of 93% (6.5 mg). Similarly, for the cationic CTAB surfactant micelles, we scaled up the synthesis of SSC-10, obtaining an isolated yield of 94% (5.4 mg). Here, we would like to mention that the solubility of the reactants, 4-(5,10,15-triphenylporphyrin-20-yl)aniline (R1) and 5-(4-carboxyphenyl)-10,15,20-(triphenyl)porphyrin (R3), is low in water, which limited the scale-up of the reactions and resulted in these product quantities. (Figure R2 and R3). Additionally, we scaled up 2D polymerization reactions (2DPI and 2DPBI) using both cationic and anionic surfactant micelles to further demonstrate the versatility of the approach. For 2D polymerization, we used porphyrin monomers 5,10,15,20-(tetra-4-aminophenyl)porphyrin (M1) and 5,10,15,20-(tetra-4-carboxyphenyl)porphyrin (M3), which have higher solubility in water due to their increased protonation sites. The isolated yields for the 2D polymers, 2DPI and 2DPBI, were 24.9 mg and 31.2 mg (92% and 94% yield), respectively. Solid-state ¹³C NMR spectroscopy confirmed the presence of imine and imidazole linkages, verifying the linkage specificity of the polymerized products (Figure R4). The isolated yield was calculated after rigorous washing (with water and THF), filtration, and subsequent drying under high vacuum. The resulting powder was then measured. In the scaled-up reactions, the isolated yield remained very consistent (Figures R1-R4). Likewise, for the 2D polymerization, the yield was comparable to the yields observed in the site-selective reactions. These results indicate that while solubility limits the scale of some reactions, the methodology maintains high selectivity and efficiency.

DLS data as the mean of three independent measurements.

a

Size d.nm	Mean Intensity Percent	Std Dev Intensity Percent	Size d.nm	Mean Intensity Percent	Std Dev Intensity Percent	Size d.nm	Mean Intensity Percent	Std Dev Intensity Percent	Size d.nm	Mean Intensity Percent	Std Dev Intensity Percent
0.4000	0.0	0.0	5.615	0.0	0.0	78.82	1.5	1.8	1195	0.2	0.3
0.4532	0.0	0.0	6.503	0.0	0.0	91.28	2.6	1.6	1281	0.0	0.1
0.5385	0.0	0.0	7.531	0.0	0.0	105.7	4.3	1.4	1484	0.0	0.0
0.6213	0.0	0.0	8.721	0.0	0.0	122.4	6.7	2.0	1718	0.0	0.0
0.7195	0.0	0.0	10.10	0.0	0.0	141.8	9.4	1.9	1990	0.0	0.0
0.8332	0.0	0.0	11.70	0.0	0.0	164.2	11.9	0.9	2305	0.0	0.0
0.9649	0.0	0.0	13.54	0.0	0.0	190.1	13.4	2.3	2689	0.0	0.0
1.117	0.0	0.0	15.69	0.0	0.0	220.2	13.2	3.9	3091	0.2	0.3
1.294	0.0	0.0	18.17	0.0	0.0	255.0	11.1	4.1	3580	0.6	1.0
1.499	0.0	0.0	21.04	0.0	0.0	295.3	7.6	2.5	4145	1.5	1.8
1.738	0.0	0.0	24.38	0.0	0.0	342.0	3.8	0.4	4801	2.7	2.7
2.010	0.0	0.0	28.21	0.0	0.0	396.1	1.2	1.4	5580	3.9	3.5
2.328	0.0	0.0	32.67	0.0	0.0	458.7	0.7	1.2	6439	0.0	0.0
2.696	0.0	0.0	37.84	0.0	0.0	531.2	0.6	1.1	7456	0.0	0.0
3.122	0.0	0.0	43.82	0.0	0.0	615.1	0.6	1.0	8635	0.0	0.0
3.615	0.0	0.0	50.75	0.0	0.0	712.4	0.6	1.0	1,000e4	0.0	0.0
4.187	0.0	0.0	58.77	0.2	0.4	825.0	0.5	0.8			
4.849	0.0	0.0	68.08	0.8	1.3	955.4	0.3	0.8			

Figure R10 | (a) Dynamic Light Scattering (DLS) analysis of micelle formation. The graph presents the size distribution of micelles, measured across three independent experiments; (b) Peak analysis of the mean values and the standard deviation.

The histogram (b) shows the mean intensity percentage as a function of the hydrodynamic diameter of the micelles (nm), with error regions indicating the standard deviation, and an average size of approximately 223 nm and a FWHM of the Biguassian fit of 177 nm. The accompanying table summarizes the mean intensity percentages and standard deviations for various particle size ranges, providing a detailed view of the size distribution. The inset image (Step-I) illustrates the micelle formation process.

Figure R11 | a) Dynamic Light Scattering (DLS) analysis of micelle formation. The graph presents the size distribution of micelles, measured across three independent experiments; b) Peak analysis of the mean values and the standard deviation.

The histogram (b) shows the mean intensity percentage as a function of the hydrodynamic diameter of the micelles (nm), with error region indicating the standard deviation, and an average size of approximately 295 nm and a FWHM of the Biguassian fit of 238 nm. The accompanying table summarizes the mean intensity percentages and standard deviations for various particle size ranges, providing a detailed view of the size distribution. The inset image (Step-II) illustrates the J-aggregated R1 adsorption on the micelles surface.

Figure R12 | a) Dynamic Light Scattering (DLS) analysis of micelle formation. The graph presents the size distribution of micelles, measured across three independent experiments; b) Peak analysis of the mean values and the standard deviation.

The histogram (b) shows the mean intensity percentage as a function of the hydrodynamic diameter of the micelles (nm), with error bars indicating the standard deviation, and an average size of approximately 1080 nm and FWHM of the peak of 919 nm. The accompanying table summarizes the mean intensity percentages and standard deviations for various particle size ranges, providing a detailed view of the size distribution. The inset image (Step-III) illustrates the bond formation and followed by multilayer adsorption.

ACTION:

Supplementary Information: Figures R10-R12 have been incorporated into the modified SI as Supplementary Fig. 13.

Reviewer #2 (Remarks to the Author):

General comment: *The authors presented a novel method to mimic on-water surface chemistry by micelles. Using the characteristics of charged surfactant molecules, a micelle structure simulating the air-water interface was formed to achieve unique reactivity and high selectivity conversion of 14 reactions. The method was extended to 2D polymerization, and the water synthesis of crystalline 2D polymer thin layer was realized. This strategy is novel and meaningful, broadening the accessibility of on-water surface chemistry in a wide range of chemical and material synthesis. However, some revision to following issues should be provided for consideration before publication.*

Response: We greatly appreciate the reviewer for the positive and insightful comments, which have been invaluable in the revision of our manuscript.

Comment 1: *In the fifth paragraph of result and discussion, “The continuous formation of imide bonds and the associated release of charges facilitated the layer-by-layer growth of a multilayer product (Fig.3f), as confirmed by a downfield ¹H NMR shift (for a detailed discussion, see Supplementary Fig. 12).” Please verify that the relevant pictures mentioned are correct.*

Response: We thank the reviewer for highlighting the need to verify the references to the figures mentioned in the manuscript. We have corrected the figure references, which should be Figure 3c and Figure 3f. In the revised manuscript, we have also added a detailed discussion in Supplementary Fig.13 regarding the layer-by-layer growth.

The reaction progression from Step-I to Step-III was thoroughly analyzed using in situ techniques such as Dynamic Light Scattering (DLS) and Nuclear Magnetic Resonance (NMR) spectroscopy, with an imide reaction (SSC-1) serving as a representative example of the layer-by-layer assembly mechanism.

Step-I: In this step, anionic micelles of SOS were formed by dissolving SOS in an aqueous solution. The ¹H NMR spectra showed characteristic peaks corresponding to the SOS surfactant within the micelles, providing a baseline for monitoring subsequent changes.

Step-II: Following the formation of micelles, protonated 4-(5,10,15-triphenylporphyrin-20-yl)aniline (R1) molecules were introduced into the aqueous solution at pH ~1.2. The ¹H NMR spectra showed a downfield shift in the characteristic peak of the SOS surfactant within 30 minutes of introducing R1, indicating increased electron density at the micelle surface due to the adsorption of the R1 molecules. This shift is attributed to the electrostatic interaction between the negatively charged head groups (SO₄⁻) of the SOS surfactant and the protonated R1 molecules. Concurrently, time-dependent DLS measurements revealed an increase in micelle size upon the introduction of R1, confirming the adsorption process. After several layers of R1 were adsorbed, no further adsorption was observed after 30 minutes, as indicated by the stabilization of the hydrodynamic diameter of the R1-SOS micelle assembly (average size 325 nm) (Figure R7a). This stability suggests that the protonated R1 molecules effectively screened the negative surface charge of the SOS micelles, leading to charge neutralization.

Step-III: The addition of naphthalene tetracarboxylic dianhydride (R2) molecules triggered a rapid increase in micelle size, as observed in time-dependent DLS measurements, which stabilized at approximately 1080 nm after 10 hours (Figure R7a). This size increase is indicative of multilayer adsorption, driven by the chemical reaction between R1 and R2 on the micelle surface, resulting in imide bond formation and the release of H₃O⁺ ions. The release of

protons from the reaction sites rendered the micelle surface negatively charged, facilitating further adsorption of positively charged R1 molecules from the bulk aqueous solution.

Figure R7 | (a) Average hydrodynamic diameter size over time measured by in-situ time-dependent DLS. (b) Elemental mapping using energy dispersive X-ray spectroscopy (EDS) in scanning transmission electron microscopy (STEM).

The negative surface charge, along with the well-defined J-aggregate template structure on the micelles, served as a driving force for the continued adsorption of R1 (Figure R7). The ongoing formation of imide bonds and subsequent charge release perpetuated the layer-by-layer growth of the multilayer site-selective product (SSC-1), which reached equilibrium after 10 hours, with a hydrodynamic diameter of approximately 1080 nm, as shown by the time-dependent DLS analysis. The observed surface charge behavior is consistent with prior studies on SOS surfactant monolayers on the water surface, including findings from ultrafast phase-sensitive interface-selective nonlinear vibrational spectroscopy, particularly sum-frequency generation (SFG) spectroscopy (*Nat Commun* **14**, 8313 (2023)).

Comment 2: The time-dependent surface tension measurements (Supplementary Fig. 2) can only indicate that the micelles still exist, but cannot explain the high stability of the micelles. And there is no clear concentration of SOS. I think there is not rigorous expression here.

Response: We thank the reviewer for highlighting the time-dependent surface tension measurements in Supplementary Figure 2. To address your concern, we also performed time-dependent DLS measurements, which demonstrated that the hydrodynamic diameter of the micelles remained constant over 30 minutes, indicating a stable micelle structure (Figure R13).

However, micelle formation above the critical micelle concentration (CMC) is inherently dynamic. To enhance the stability of the micelles, we introduced porphyrin molecules (R1/R3) immediately after the formation of micelles in Step-II. This approach helps stabilize the micelle structure by reducing their dynamic nature, thereby maintaining structural integrity for subsequent reactions.

Figure R13 | DLS measurements of SOS micelles at 5 minutes and 30 minutes, showing consistent hydrodynamic diameter over time, indicating stable micelle structure.

Regarding the concentration of SOS used in the experiments, the concentration details were provided in the Methods section. For clarity, we include this information in the main text: “Micelles of sodium oleyl sulfate (SOS), an anionic surfactant, were prepared by dissolving 20 mg of SOS (2.87 mmol/L) in an aqueous solution above its critical micelle concentration (CMC) of 1.7 mmol/L. This solution was placed in a 100 mL Duran glass bottle with a GL-45 cap.”

ACTION:

Supplementary Information: Figures R13 has been incorporated into the modified SI as Supplementary Fig. 2.

Comment 3: *The micelle surface head group promotes the formation of R1 molecules within a J-aggregated structure. It would be better to add the case with or without micelles in Fig. 2d as a comparison.*

Response: We appreciate the reviewer’s suggestion to include a comparison of the J-aggregation of R1 molecules with and without micelles in Figure 2d. In response, we conducted additional experiments using PXRD to provide a clearer comparison of the aggregation behavior above and below the critical micelle concentration (CMC).

PXRD analysis was performed to compare the structural organization of R1 molecules above and below the CMC of the SOS surfactant. Above the CMC (with micelle formation), we observed a strong J-aggregated structure characterized by a smaller unit cell, indicating enhanced packing and order facilitated by the micelle surface. However, below the CMC (without micelles), we did not observe any J-aggregated structure. Instead, the PXRD revealed a distinctly different structure, indicating a larger unit cell and a lack of the self-assembled ordered structure. This result highlights the critical role of micelles in promoting the unique J-aggregated structure.

Figure R14 | PXRD analysis of R1 molecules above (blue) and below (gray) the CMC of SOS surfactant. Above the CMC, J-aggregation with a smaller unit cell is observed, while below the CMC, a different structure with a larger unit cell and no J-aggregation is seen, highlighting the role of micelles in promoting J-aggregation.

ACTION:

Supplementary Information: Figure R14 has been incorporated into the modified SI as Supplementary Fig. 5.

Comment 4: *In the part of extension to 2D polymerization on the micelle surface, the author mentions the (A₄+B₂)-type monomers, and its related information is not found in the manuscript, which I think should be added.*

Response: We thank the reviewer for pointing out the need for additional information regarding the (A₄+B₂)-type monomers used in the extension to 2D polymerization on the micelle surface. We would like to clarify that the monomers M1, M2, M3, and M4 used in our study were commercially sourced. Specifically, M1 and M3 were purchased from PorphyChem, M2 was obtained from TCI, and M4 was acquired from Sigma-Aldrich. All monomers were used directly without further purification.

Figure R15 | Schematic representation of 2D polymerization on micelle surfaces using (A₄+B₂)-type monomers. Top: Synthesis of 2DPI with monomers M1 and M2 on the anionic SOS micelle surface. Bottom: Synthesis of 2DPBI with monomers M3 and M4 on the cationic CTAB micelle surface.

ACTION:

Supplementary Information: Additional details regarding the (A₄+B₂)-type monomers have been incorporated into the revised SI under the Materials and Methods section.

Comment 5: *In the sequential assembly process on micellar surfaces, the size of micelles plays a crucial role in achieving highly crystalline, large-area circular sheets and few-layer stacked 2D polymers. However, there is no discussion on the effect of adjusting micelles size on the formation of 2D polymers in the current characterization, which needs to be clarified in the revision.*

Response: We appreciate the reviewer's insights and concerns. The interaction between porphyrins and surfactants, along with the effect of surfactant charge, plays a crucial role in determining the overall size and structure of the 2D polymers. Below, we provide a more detailed explanation of these interactions and how micelle size and surface charge contribute to 2D polymer synthesis.

Influence of surface charge density and micelle size on 2D polymer synthesis:

1. **Surface charge density:** The surface charge density of micelles is a critical factor influencing the adsorption and assembly of porphyrin molecules on the micelle surface. The charge density follows a double-layer model, where the charged surfactant head groups interact with oppositely charged porphyrins, facilitating their adsorption onto the micelle surface. High surface charge density enhances the electrostatic interactions, promoting the formation of ordered assemblies and ultimately influencing the size and crystallinity of the resulting 2D polymers.

2. **Impact of micelle size:** The curvature of the micelle surface, particularly angular curvature, significantly influences the packing and arrangement of porphyrin molecules. Larger micelles have lower curvature, providing a more planar surface, which allows for a more even distribution with less distortion, better organization of the porphyrin self-assembled structure, and stable adsorption of porphyrins. This structured assembly is essential for developing 2D polymers with larger domain sizes and higher crystallinity. In our study, larger micelles, such as those formed by the M1-SOS anionic surfactant (~318 nm), facilitate the pre-assembly of monomers over more extensive domain areas, promoting higher crystallinity in 2DPI. In contrast, smaller micelles, such as those formed by the M3-CTAB cationic surfactant (~106 nm), are constrained by their limited surface area and increased angular curvature, leading to reduced crystallinity and a tendency towards polycrystallinity with smaller circular-sheet domains in 2DPBI.

ACTION:

Supplementary Information: The discussion on the influence of micelle size on 2D polymer synthesis is included in Supplementary Fig. 42.

Comment 6: *Why choose SOS and CTAB instead of other charged surfactants? Can other charged surfactants form similar 2D polymer sheets?*

Response: We appreciate the reviewer's comment. Our selection of SOS (anionic) and CTAB (cationic) was based on our group's prior experience with surfactant-monolayer-assisted interfacial synthesis (SMAIS). Both SOS and CTAB have consistently shown the ability to form crystalline soft template pre-assembled structures, facilitating the formation of highly crystalline, large-area 2D polymer sheets. This made them ideal candidates for validating our new approach in this work.

We are currently expanding this strategy to explore other surfactants, including mixed surfactants and Gemini surfactants, which can form larger micelles (above tens of microns) and various shapes such as ellipsoidal, cylindrical, and unilamellar vesicles. This ongoing work aims to create micelles with larger accessible surface areas, potentially enabling the growth of single-crystalline 2D polymers. The results of these experiments and their implications for broader surfactant selection will be published in a separate report at a later stage.

Comment 7: *Some expressions need to be added with references, such as the sixth paragraph of result and discussion, "imides typically require high temperatures (>170°C) in bulk organic reactions".*

Response: We thank the reviewer for pointing out the need for additional references. We have added the appropriate reference (*Nat. Commun.* 5, 4503,5503 (2014)) in the sixth paragraph of the Results and Discussion section to support the statement that "imides typically require high temperatures (>170°C) in bulk organic reactions."

Comment 8: *Adding statistics on the size of 2DPI and 2DPBI in Figures 4b and 4e will be more complete.*

Response: We appreciate the reviewer's suggestion. Thus we have added the size distribution histograms for both 2DPI and 2DPBI. For 2DPI, the average size is approximately 2.25 μm with a standard deviation of 0.40 μm , and for 2DPBI, the average size is about 1.14 μm with a standard deviation of 0.30 μm .

Figure R16 | Size distribution histograms of 2DPI (left) and 2DPBI (right). The average size of 2DPI is approximately 2.25 μm , while the average size of 2DPBI is about 1.14 μm .

ACTION:

Supplementary Information: Figure R16 has been incorporated into the modified SI as Supplementary Fig. 39.

Reviewer #3 (Remarks to the Author):

General comment: *In this manuscript, A. Prasoon et al. presented a new approach that mimics on-water surface chemistry using micelles. The steps of this approach include the formation of charged surfactant molecules beyond their CMC, assembly of one type of reaction molecule, and chemical reactions taking place on the surfaces of micelles. This approach created an environment where the hydrophobic core and surrounding water layer form an interface that facilitates chemical reactions akin to those observed in on-water surface chemistry. However, there are some scientific and technical issues that prevent this reviewer from recommending acceptance of the manuscript for publication.*

Response: We appreciate the reviewer for the valuable comments, which have been very helpful in the revision of our manuscript.

Comment 1: *The amino-substituted porphyrin is a 2D molecule. On the surface of the micelles, the molecules will be parallel or perpendicular to the surfaces of micelles?*

Response: We appreciate the reviewer's comment. The porphyrin molecules, with their protonated core, interact strongly with the SO_4^- head groups of the SOS surfactant, aligning parallel to the micelle surface with a slight tilt due to J-aggregation, as confirmed by PXRD analysis (Figure 2d). This arrangement facilitates optimal electrostatic interactions and stabilizes the J-aggregated structure with a minimal slipping angle. To clarify this orientation, we have added a schematic illustrating the assembly of the porphyrin molecules from both top and side views on the micelle surface.

Figure R17 | Schematic illustration of the assembly of R1 porphyrin molecules on SOS micelle surfaces. Left: R1 molecules are shown aligning parallel to the surface of SOS micelles, interacting via electrostatic attraction with the SO_4^- head groups. Right: Top and side views depict the J-aggregated structure with a short slipping distance.

Comment 2: *The authors claimed that the 2DPI and 2DPBI were highly crystalline structured. What about the polymerization degree? Crystallinity of the polymers is highly related to their polymerization degrees and should be compared with that of the materials obtained under a regular way.*

Response: We appreciate the reviewer's insightful comment regarding the relationship between the crystallinity of 2DPI and 2DPBI and their degree of polymerization (DP). While the DP is an important factor influencing crystallinity, directly measuring or quantifying DP in 2D polymers and related materials, such as 2D covalent organic frameworks (2D-COFs), is challenging due to their extended 2D network structures. Conventional methods for

determining DP of linear polymers, such as Gel Permeation Chromatography (GPC), are not applicable to 2D polymers and 2D-COFs because these materials are typically insoluble and infusible. As a result, direct measurement of molecular weight, and consequently DP, is not feasible for these systems.

Indirect estimation of DP via crystallinity and structural characterization:

1. Structural characterization: In 2D polymers and 2D-COFs, crystallinity is typically assessed through techniques like Powder X-ray Diffraction (PXRD) and Transmission Electron Microscopy (TEM), which provide insights into the ordered arrangement and periodicity of the frameworks. Sharp diffraction peaks in PXRD and uniform sheet-like structure in TEM images are indicative of high crystallinity, which indirectly suggests a substantial degree of polymerization. However, defects, irregularities, and amorphous regions that can occur during synthesis complicate the direct estimation of DP.

2. Spectroscopic methods: Spectroscopic techniques, such as Infrared (IR) and Raman spectroscopy, can also offer insights into the extent of polymerization by monitoring the disappearance of monomer-specific peaks and the appearance of new polymer-specific bands. However, these methods provide indirect evidence and do not quantify DP directly.

In the literature, the degree of polymerization in 2D polymers and 2D COFs is often inferred indirectly through crystallinity. Highly crystalline 2D polymers and 2D COFs are generally assumed to have high degrees of polymerization due to their well-ordered frameworks. Similarly, the high crystallinity observed in 2DPI and 2DPBI in our study, as demonstrated by PXRD and TEM images showing an average crystal domain size of 40 nm, along with IR spectroscopic evidence showing the disappearance of monomer-specific peaks (NH_2 for 2DPI, COOH for 2DPBI) and the appearance of new polymer-specific bands (imine for 2DPI, benzimidazole for 2DPBI), suggests a significant degree of polymerization.

Comment 3: *About Figure 1a, the left and middle images of Figure 1a have nothing to do with this research. Figure 1 should be simplified. If any reactions or procedures are not related to this research, they should be removed.*

Response: We appreciate the reviewer's suggestion. However, we respectfully hold a different perspective. In our current manuscript, we are introducing a new approach that mimics on-water surface chemistry using micelles. The inclusion of the left and middle images in Figure 1a is intended to provide a comparative schematic of traditional on-water surface chemistry alongside our new approach. These images are not only relevant but also crucial for illustrating how our method parallels and extends the principles of established on-water surface chemistry techniques. We believe this comparison enhances the understanding of the significance of our current work and its connection to existing methods.

Comment 4: *The inset of Figure 2a shows an SEM image, which is not clear and lacks a scale bar. TEM should give clearer images. The images obtained before and after porphyrin assembly should be compared.*

Response: Actually, the direct imaging of micelles under FESEM or TEM presents significant challenges due to their dynamic nature and instability outside of an aqueous environment. Without stabilization, micelles alone cannot be effectively visualized because they do not maintain their structure under electron microscopy conditions.

In step II, we observed that the hydrodynamic diameter of the micelles increased from an average size of 223 nm (Step I) to approximately 295 nm after porphyrin adsorption, indicating that only a few layers of porphyrins are adsorbed onto the micelle surface. Due to this minimal layer formation, imaging under TEM remains difficult as the structure cannot withstand the irradiation from an electron beam. Consequently, we employed FESEM with a very short scan duration to minimize damage, which unfortunately affected the image quality as the sample started to degrade under electron irradiation. We apologize for the oversight regarding the scale bar. To enhance clarity, we have now included a scale bar for reference in figure 2a main text.

Comment 5: *The authors used UV-visible spectroscopy to confirm the formation of J-aggregates. The absorption of porphyrin is very sensitive to the solution circumstance. An absorption spectrum of porphyrin should be obtained at a SDS concentration lower than the CMC.*

Response: We appreciate the reviewer's comment. In response, we performed UV-visible spectroscopy on R1 both above and below the CMC of SOS surfactant. As shown in Figure R18, at concentrations above the CMC, where micelles are present, R1 exhibits a pronounced red shift in the Soret band along with a sharper absorption peak. This red shift is indicative of J-aggregate formation for R1. Conversely, at SOS concentrations below the CMC, the UV-visible absorption spectrum of R1 is significantly broadened, and the characteristic red shift of the Soret band is absent. This broadening suggests that the porphyrin molecules are not forming ordered J-aggregates. Instead, the broad spectrum below the CMC indicates a different, less ordered aggregation or individual porphyrin molecules in solution. This comparison clearly demonstrates that the micelle surface (template) is crucial for the pre-assembly of R1.

Figure R18 | UV-visible absorption spectra of R1 porphyrin in the presence and absence of SOS surfactant at concentrations above and below the critical micelle concentration (CMC). The R1 spectrum without SOS surfactant serves as a reference.

ACTION:

Supplementary Information: Figure R18 has been incorporated into the modified SI as Supplementary Fig. 6.

Main Text: We have added the following sentence to the main text (page 6): “As a control experiment, we performed UV-visible spectroscopy and PXRD experiments on R1 both above and below the CMC of SOS surfactant, demonstrating that J-aggregate formation occurs only above the CMC, where micelles are present (Supplementary Fig.5 and 6).”

Comment 6: THF was used to wash the porphyrin J-aggregates. What about the solubility of the porphyrin in THF?

Response: To selectively remove the SOS micelles without disrupting the J-aggregated porphyrins, we used cold THF (approximately -20°C) for washing the aggregates on filter paper. Cold THF interacts preferentially with the SOS surfactant due to its polar nature, which disrupts the hydrophobic interactions stabilizing the micelles, leading to their disassembly. At lower temperatures, THF has reduced solvating power, which minimizes the dissolution of the J-aggregated porphyrins. The J-aggregates are stabilized by strong π - π and polarized- π interactions that are less affected by cold THF.

We performed the washing with minimal amounts of THF, ensuring that the J-aggregates remained intact. Initially, the R1-SOS assembly formed spherical structures with a hydrodynamic diameter of approximately 300 nm. After washing, the removal of the SOS micelles transformed the morphology into a circular sheet-like structure, similar to how a balloon flattens when deflated, as confirmed by FESEM images (Figure R19).

Figure R19 | FE-SEM images reveal circular sheet-like structures of the SOS-R1 (after washing with THF)

Comment 7: XRD data were used to confirm the formation of porphyrin J-aggregates (1.346 and 1.020 nm vs. 1.66 and 1.48 nm). How were the values of 1.66 and 1.48 nm obtained? Measurement should be carried out with control samples, which was obtained at a SDS concentration lower than the CMC.

Response: We appreciate the reviewer's valuable comment. To clarify, these values represent the reference dimensions of individual R1 porphyrin molecules, measured using Mercury 3.10.1 software, and serve as a baseline for comparison with the J-aggregated structures. The unit cell parameters of the J-aggregates formed on the micelle surface are significantly reduced compared to those of the individual R1 molecules, with observed lattice parameters of approximately 1.34 nm and 1.02 nm. This reduction in lattice size indicates enhanced packing and ordering of the porphyrins within the J-aggregates, characterized by a short slipping distance.

To further address the reviewer's concern, we conducted additional PXRD experiments to compare the aggregation behavior of R1 molecules above and below the critical micelle concentration (CMC) of the SOS surfactant. Above the CMC, where micelles are present, the PXRD data confirmed the formation of a strongly J-aggregated structure with a smaller unit cell. Conversely, below the CMC, where micelles do not form, the PXRD analysis revealed a distinctly different and less ordered structure with a larger unit cell, indicating the absence of J-aggregation.

Figure R20 | Structural analysis of R1 assembly on micelle surfaces using PXRD. (a) Molecular structure of an individual R1 molecule. (b) Lattice structure of the pre-assembled J-aggregate on the micelle surface. (c) PXRD analysis of R1 molecules above (blue) and below (gray) the CMC of SOS surfactant. Above the CMC, J-aggregation is observed with a smaller unit cell, while below the CMC, a different structure with a larger unit cell and no J-aggregation is present, highlighting the role of micelles in promoting J-aggregation.

ACTION:

Supplementary Information: Figure R20 has been incorporated into the modified SI as Supplementary Fig. 5.

Comment 8: *The color change may not due to the reaction. Porphyrin shows different color at alkaline or acid media. A control test should be performed with only R2.*

Response: We appreciate the reviewer's valuable comment. It is true that porphyrin molecules exhibit different colors in alkaline or acidic media, and we acknowledge that this could influence the observed color change.

Our sequential three-step reaction is carefully controlled under specific pH conditions, and the observed color change upon adding R2 in Step-III serves as a primary indication that a reaction is occurring. Specifically, when the naphthalene tetracarboxylic dianhydride (R2) was introduced into the aqueous solution, we observed a rapid color change from light green to dark orange within a minute. While this color change alone is not conclusive evidence of the reaction, it suggests that a chemical transformation may be taking place.

To confirm the formation of the SSC-1 product, we conducted comprehensive analyses using MALDI-TOF MS, NMR, and FTIR spectroscopy, which collectively provided strong evidence of the successful imide bond formation between R1 and R2. As a control experiment, we considered testing R2 alone; however, R2 molecules do not dissolve in water and require alkaline media to dissolve. Therefore, a control experiment with only R2 in the same reaction conditions was not feasible.

Comment 9: *On page 6, "FE-SEM revealed a spherical morphology with an average diameter of ~1 μm and a thickness of ~18 nm". It is hard to draw such conclusion.*

Response: We apologize for the oversight and thank the reviewer for pointing this out. The spherical morphology was observed before washing with THF. After washing with THF to remove the micelles, the morphology is correctly described as a circular sheet structure. This

circular morphology was confirmed through AFM analysis, which measured the thickness of these sheets to be approximately 18 nm.

ACTION:

Main Text (page 6): We have added the following sentence to the main text: “Atomic force microscopy (AFM) analysis confirmed the presence of circular sheet structures with a thickness of approximately 18 nm.”

Comment 10: *The SEM image in Figure 3a is not clear and lacks a scale bar.*

Response: We appreciate the reviewer’s valuable comment. To provide a clearer representation, we have conducted additional TEM imaging, which offers improved resolution and detail of the structures. The TEM images have been added to the revised manuscript, and the scale bar has been included in Figure 3a.

Figure R21 | TEM image showing the spherical morphology of the SOS-R1-R3 assembled structure before washing with THF.

ACTION:

Main text: In the revised manuscript, we have updated Figure 3d with an additional TEM image.

Comment 11: *Line 195 on page 8, “which stabilized at 1020 nm after 5 hours”. However, the data in Figure 3a do not show 1020 nm.*

Response: We appreciate the reviewer’s valuable comment. We apologize for the oversight and any confusion caused. The correct statement is that the hydrodynamic size stabilized at an average diameter of 1080 nm after 10 hours.

To clarify, in the previous version of the manuscript, Figure 3a presented data for the first 5 hours of the reaction, during which the stabilization at 1080 nm had not yet been observed. The stabilization is shown in Figure 3b, which provides data for up to 10 hours. In the revised manuscript, we have updated Figure 3a to include data points for up to 10 hours. Additionally, we conducted further DLS measurements after 10 hours of adding R2, with the results averaged

from three independent measurements, including error bars representing the standard deviation. This additional data confirms the stabilization of the hydrodynamic size at approximately 1080 nm.

Figure R12 | a) Dynamic Light Scattering (DLS) analysis of micelle formation. The graph presents the size distribution of micelles, measured across three independent experiments; b) Peak analysis of the mean values and the standard deviation.

The histogram (b) shows the mean intensity percentage as a function of the hydrodynamic diameter of the micelles (nm), with error bars indicating the standard deviation, and an average size of approximately 1080 nm and FWHM of the peak of 919 nm. The accompanying table summarizes the mean intensity percentages and standard deviations for various particle size ranges, providing a detailed view of the size distribution. The inset image (Step-III) illustrates the bond formation and followed by multilayer adsorption.

ACTION:

Main text: In the revised manuscript, we have updated Figure 3a to include data points for up to 10 hours

Comment 12: *No substantial evidences supported the formation of a hollow structure with a hydrophobic core surrounded by water. The authors are advised to conduct further investigations into this structure.*

Response: We appreciate the reviewer's constructive comments. Thus we employed confocal fluorescence microscopy to directly observe the fluorescence emitted from the SSC-1 on the micelle surface in an aqueous solution, thus assessing the presence of hollow micelles and the localization of the assemblies.

Upon completing Step-III, where the SSC-1 product formed on the micelle surface, the hollow micelles displayed bright fluorescence along their surface boundaries under the microscope. By exciting the sample at 552 nm, corresponding to the SSC-1 excitation, we captured and analyzed the emitted light. Adjusting the microscope's focal plane (Figure R22, panels a-c) enabled visualization of fluorescence at the micelle edges, while dark areas indicated regions not illuminated by the fluorescent laser. In the micelle marked with a red circle, adjusting the focal plane revealed high fluorescence at the edges but none in the central black region, confirming the hollow micelle core lacks reactant molecules (R1 or R2). We also recorded a video showcasing the dynamic behavior and structural integrity of the micelles. Quantitative analysis of fluorescence intensity (Figure R22-d) estimated the SSC-1 distribution on the micelle surface. Gray-scale visualization further confirmed no fluorescence inside the micelles, indicating the SSC-1 product is located only on the surface.

Figure R22 | Sequences (a, b, c) of the emitted fluorescence at different focal levels. The black color indicates the hollow core of the micelles and the absence of the SSC-1 product inside. (d) Gray-scale visualization of the emitted fluorescence intensity upon laser excitation, showing no light intensity recorded in the hollow body of the micelles. (e) Elemental mapping using energy dispersive X-ray spectroscopy (EDS) in scanning transmission electron microscopy (STEM).

ACTION:

Supplementary Information: Figure R22 (a-d) has been incorporated into the modified SI as Supplementary Fig. 7.

Comment 13: Line 197 on page 8, “releasing H_3O^+ ions”. R2 is an anhydride, which is sensitive to water. Why did it not react with water when it added into the reaction mixture? Suppose R2 does not react with water, it is H_2O that is released after the reaction between R1 and R2. Where is the proton from? R2 has two functional groups, why did the other functional group of R2 not react with another R1?

Response: We appreciate the reviewer’s thoughtful questions. To clarify, here is a detailed explanation of the reaction process and the origin of the protons:

In SSC-1 reaction, R1 molecules are used in their protonated form at pH ~ 1.2 , which creates a highly acidic environment conducive to the reaction with R2, and it should be mentioned that R2 is dissolved in a basic solution using LiOH before being introduced into the reaction beaker. When protonated R1 (NH_3^+) interacts with R2 on the micelle surface, the proton from the NH_3^+ group is released when the imide bond forms between R1 and R2.

Site-selective reaction:

The selective reaction of only one anhydride group of R2 is influenced by the constrained geometry and polarized- π interactions of the J-aggregated R1 molecules on the micelle surface, which restrict the movement or rotation of the J-aggregated R1. This constrained arrangement of the porphyrin molecules facilitates the directional alignment of the R2 reagent, resulting in the selective formation of a one-sided imide product.

Figure R8 | A schematic illustrating a site-selective chemical reaction on the surface of SOS micelles.

ACTION:

Supplementary Information: Figure R8 has been incorporated into the modified SI as Supplementary Fig. 12.

Comment 14: In the model reaction, only one functional group of R2 reacted with R1. Why could the polymerization take place?

Response: In the site-selective model reaction, we specifically used R1 molecules, which contain only one amino group, leading to the formation of a one-sided imide product. For the 2D polymerization reaction, however, we utilized a different monomer, M1, which possesses four amino groups.

Figure R23 | Porphyrin molecule R1 used for site-selective reaction and M1 used for 2D polymerization.

Comment 15: *Figure 4b and 4e only show particles, lacking direct evidences for sheet-like morphologies of the products.*

Response: We appreciate the reviewer's valuable comment. After the 2D polymerization reaction, we performed a washing step with cold THF to remove the SOS micelles. This washing process led to a transformation in morphology, where the initially spherical micelle-associated structures flattened into circular sheet-like forms. The FESEM analysis shows that the spherical structures have indeed flattened, suggesting the formation of circular sheet-like structures.

Figure R24 | FE-SEM image reveal circular sheet-like structures of the 2DPI (after washing with THF)

Comment 16: *Scale bars should be given in the AFM images. They look like particles. Cross-section analysis is needed.*

Response: We have included scale bars in the revised AFM images to provide a clear size reference. Additionally, AFM analysis revealed that the 2D circular sheets have a thickness of approximately 18.4 nm.

Figure R25 | AFM image reveal circular sheet-like structures of the 2DPBI (after washing with THF)

ACTION:

Supplementary Information: Figure R25 has been incorporated into the modified SI as Supplementary Fig. 40.

Comment 17: *From Line 264 to 270 on page 10, evidences for achieving the cell parameters are not enough. In Figure 4c, the angle between the different directions of lattices is not 90 degrees.*

Response: We appreciate the reviewer's valuable comment. In response, we would like to clarify that the corresponding fast Fourier transform (FFT) image of Figure 4c shows that the angle between the points is indeed 90 degrees, confirming the expected orthogonal arrangement. However, it is important to note that the circular curvature of the micelles can reduce crystallinity and often leads to a polycrystalline nature in the produced 2D polymers, with an average crystal domain size of 40 nm. This curvature influences the crystal domains, causing them to orient in different directions due to the angular nature of the micelle template. This explains the observed variations in lattice orientation, as seen in Figure R26.

Figure R26 | TEM image showing different alignments of crystal domains in the 2D polymer (2DPI). The observed variations in domain orientation are influenced by the angular curvature of the micelle template, which can reduce crystallinity and lead to a polycrystalline nature with domains aligned in multiple directions.

ACTION:

Supplementary Information: Figure R26 has been incorporated into the modified SI as Supplementary Fig. 42.

Comment 18: *The particles shown in Figure 4b and 4e were not uniform in size. Distribution analysis may be required.*

Response: Following your suggestion, we have added the size distribution histograms for both 2DPI and 2DPBI. For 2DPI, the average size is approximately 2.25 μm with a standard deviation of 0.40 μm , and for 2DPBI, the average size is about 1.14 μm with a standard deviation of 0.30 μm .

Figure R8 | Size distribution histograms of 2DPI (left) and 2DPBI (right). The average size of 2DPI is approximately 2.25 μm , while the average size of 2DPBI is about 1.14 μm .

ACTION:

Supplementary Information: Figure R8 has been incorporated into the modified SI as Supplementary Fig. 39.

Comment 19: *Can the authors obtain the number of layers in the stacked 2D polymers?*

Response: Actually, accurately determining the exact number of layers is challenging due to the polycrystalline nature of the sample. Nevertheless, following your suggestion, we provide here a rough estimation based on the overall thickness of the 2D polymer, which is approximately 18 nm. Given that the interlayer π - π stacking distance in the 2D polymer is around 0.4 nm, we estimate that the number of stacked layers ranges from 45 to 50.

Comment 20: *In figure S32a, where is the material, dark area or light area?*

Response: In the image, the darker areas represent the 2DPI circular sheets. The lighter areas correspond to small particles formed during vigorous washing with THF.

Detailed response to the comments from the reviewers

Reviewer #1 (Remarks to the Author):

General comment: *The authors have answered my initial concerns and made the appropriate changes. I would suggest publication at this stage.*

Response: We greatly appreciate the reviewer for the highly positive comments with recommendation for publication.

Reviewer #2 (Remarks to the Author):

General comment: *The manuscript has been well revised and could be accept. Following the revisions, the discussion is more comprehensive, and the experimental approach is more robust than previously. The novelty of this research is significant, and it is expected to attract the interest of readers of Nature Communications. It is hoped that the author can successfully extend this strategy to other surfactants in the future to achieve more in-depth research.*

Response: We greatly appreciate the reviewer for the highly positive comments with recommendation for publication.

Reviewer #3 (Remarks to the Author):

General comment: *The authors have modified the manuscript following the comments raised in the first round of review. While most of the comments have been reasonably responded and been reflected in the revised manuscript, there are still a couple of concerns.*

Response: We greatly appreciate the reviewer for the highly positive comments.

Comment 1: *The authors explained how difficult to measure the polymerization degree of a 2D polymer, which is acceptable. What about the crystallinity of their material when compared with the material obtained under a regular way?*

Response: We appreciate the reviewer's valuable comment. Here a comparative analysis was performed to compare the crystallinity of 2D polymers using powder X-ray diffraction (PXRD) and transmission electron microscopy (TEM). Specifically, 2DPI and 2DPBI were synthesized using both a micelle-assisted approach and the conventional solvothermal bulk organic synthesis commonly reported in the literature.

PXRD analysis revealed that the Full Width at Half Maximum (FWHM) values for the (100) plane of the micelle-assisted synthesis of 2DPI and 2DPBI were 0.25° and 0.36° , respectively. In contrast, the solvothermal synthesis approach showed broader peaks with FWHM values of 0.57° for 2DPI and 0.83° for 2DPBI, indicating lower crystallinity, as higher FWHM values correspond to smaller crystallite sizes and less ordered structures.

Figure R1 | (a) PXRD patterns comparing the crystallinity of 2D polymers synthesized by micelle-assisted and solvothermal methods. The sharper peaks observed in the micelle-assisted synthesis indicate higher crystallinity. (b) Comparison of FWHM values for the (100) plane reflections, showing lower FWHM for micelle-assisted synthesis (bars 1 and 3) and higher FWHM for solvothermal synthesis (bars 2 and 4).

Figure R2 | (a) TEM images comparing the crystallinity of 2D polymers synthesized by micelle-assisted and solvothermal methods. Images (a) and (d) represent 2DPI and 2DPBI synthesized via the micelle-assisted approach, showing well-defined, continuous lattice fringes over extended areas. In contrast, images (b), (c), (e), and (f) show 2DPI and 2DPBI synthesized via the solvothermal approach, exhibiting less distinct lattice patterns with noticeable disorder, indicative of smaller crystalline domains and lower overall crystallinity.

Additionally, TEM imaging showed that the micelle-assisted synthesis of 2DPI and 2DPBI exhibited well-defined and continuous lattice fringes over extended areas (Figure R2a and R2d), characteristic of high crystallinity and long-range order. In contrast, the 2DPs (2DPI and 2DPBI) synthesized via the solvothermal approach displayed less distinct lattice patterns with noticeable disorders (Figure R2b-c and R2d-e), indicative of smaller crystalline domains and lower overall crystallinity.

ACTION:

Supplementary Information: Figures R1 and R2 have been incorporated into the revised Supplementary Information as Supplementary Figures 41 and 42, respectively. Details of the synthesis of 2DPI and 2DPBI via the solvothermal approach are included in the methods section of the Supplementary Information.

Comment 2: *Figure 2d shows the PXRD data of the R1 assembled structure. The indexing may be wrong because (100) was used for indexing two different peaks. Please check them carefully.*

Response: We apologize for the oversight and thank the reviewer for pointing it out. The correct indexing is (110) instead of (100), and this has been corrected in the updated figure.

Detailed response to the comments from the reviewers

Reviewer #3 (Remarks to the Author):

General comment: *I am satisfied with the revisions made by the authors. Therefore, I recommend acceptance of this version of the manuscript for publication.*

Response: We greatly appreciate the reviewer for the highly positive comments with recommendation for publication.